# RESOLVING LABEL UNCERTAINTY
# WITH IMPLICIT GENERATIVE MODELS

## ABSTRACT

In prediction problems, coarse and imprecise sources of input can provide rich information about labels, but are not readily used by discriminative learners. In this work, we propose a method for jointly inferring labels across a collection of data samples, where each sample consists of an observation and a *prior belief* about the label. By implicitly assuming the existence of a generative model for which a differentiable predictor is the posterior, we derive a training objective that allows learning under weak beliefs. This formulation unifies various machine learning settings: the weak beliefs can come in the form of noisy or incomplete labels, likelihoods given by a different prediction mechanism on auxiliary input, or common-sense priors reflecting knowledge about the structure of the problem at hand. We demonstrate the proposed algorithms on diverse problems: classification with negative training examples, learning from rankings, weakly and self-supervised aerial imagery segmentation, co-segmentation of video frames, and coarsely supervised text classification.

## 1 INTRODUCTION

We consider the problem of *joint* inference of latent label variables $\ell_i$ in a collection of data samples indexed by $i$ consisting of observations (features) $x_i$ and corresponding *prior beliefs* about their latent label variables $p_i(\ell)$.

Two illustrative examples are shown in Fig. 1. In the first example, the $x_i$ are 784-dimensional vectors representing 28×28 MNIST digits. We aim to infer the digit classes $\ell_i \in \{0, 1, ..., 9\}$ for all images in the given collection based on data in which we are given just one *negative* label per sample, i.e., the prior beliefs $p_i(\ell)$ (top row) are uniform over all classes except for one incorrect class. The procedure described in this paper produces inferred distributions over labels (bottom row) that are usually peaky and place the maximum at the correct digit 97% of the time (Fig. 3).

In the second example, the observations $x_i$ are image patches centered around each pixel coordinate $i$ in a Surrealist painting, with patch size (11×11) equal to the receptive field of a 5-layer convolutional neural network used in our inference procedure. The prior beliefs $p_i(\ell)$ are distributions over 3 classes (sky, boat, water) depending on the coordinate $i$. The joint inference of all labels in this image yields a feasible segmentation despite the high similarity in colors and textures (see §E.4).

In both examples, the inference technique needs to estimate statistical links between observations $x_i$ and corresponding latents $\ell_i$ that would both be highly confident (i.e., lead to low entropy in the posterior distributions) and explain the varying prior beliefs, which typically have low confidence (high entropy in the prior distributions). This problem of training on weak beliefs, in some form, is often encountered in machine learning, e.g, weak supervision and semi-supervised learning, domain transfer, and integration of modalities – settings where coarse, partial, or inexact sources of data can provide rich information about the state of a prediction instance, though not always a "ground truth" label for each instance.

In many such settings, fusing the weak input data into a probability distribution over classes is a more natural alternative to transforming the weak input into hard labels [26]. However, because supervised models target the distribution over labels, training machine learning models with supervision from probabilistic prior beliefs results in uncertain predictions. Most approaches to resolve these uncertainties involve iterative generation of hard pseudo labels [56], or loss functions promot-

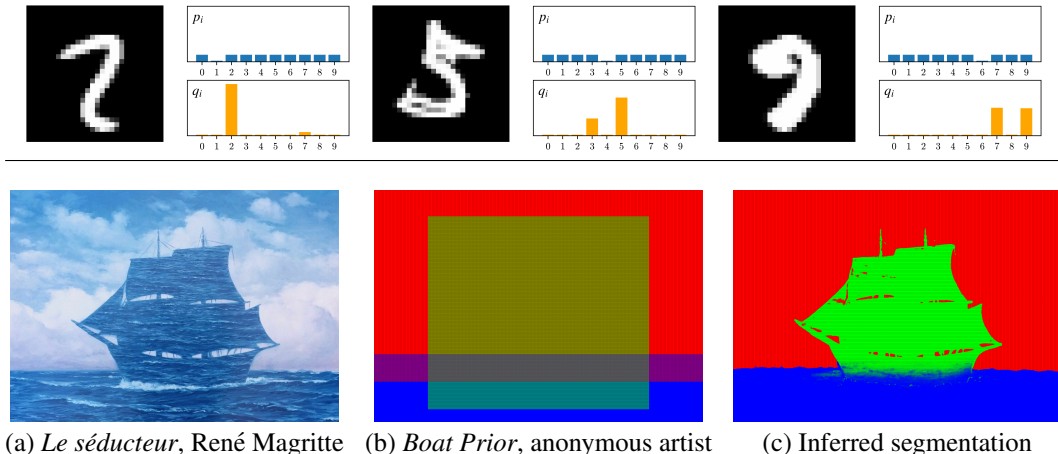

(a) *Le séducteur*, René Magritte  (b) *Boat Prior*, anonymous artist  (c) Inferred segmentation

Figure 1: **Above:** Inference of latent MNIST digit classes with negative label supervision using a small CNN trained on the **RQ** criterion (§2.1). **Below:** Joint inference of latent pixel classes in an image (a). The prior beliefs $p_i(\ell)$ over three classes – sky (red), boat (green), water (blue) – are manually set (b). A small CNN using no data except (a,b) infers the posterior classes (c).

ing low entropy of predictions [35; 55; 59; 54]. Typically, these approaches are application-specific [10; 57; 1; 22]. Further discussion of related work is provided in §B and §C.

Our key modeling insight is to associate the output distribution of a discriminative model, a feed-forward neural network $q$, with an *implicit generative model* (§2.1) of features and to consider the given prior belief as part of that model. We show that without estimating the full generative model, it is possible to learn the network that performs inference in it and reap the benefits of generative modeling, including high certainty in the posterior under soft priors and rich opportunities to model structure in the prior beliefs. We validate the effectiveness of our approach with experiments (§4, §E) that highlight: prior beliefs as a natural way to fuse weak inputs, graceful degradation of performance with increasingly noisy or incomplete inputs, and comparison of our implicit generative model with explicitly generative modeling approaches.

## 2 BACKGROUND AND APPROACH

**Supervised learning on prior beliefs.** Supervised learning models, including many neural nets, are typically trained to minimize the cross-entropy $-\sum_i \sum_\ell p_d^i(\ell) \log q_i(\ell)$ between the data distribution over labels $p_i^d(\ell)$ and the distribution $q_i(\ell) = q(\ell|x_i; \theta)$ output by a predictor $q$ using data features $x_i$. This is equivalent to minimizing the KL divergence $\sum_i \mathrm{KL}(p_i^d \| q_i)$ which is not well-suited to training with uncertain labels defined by prior beliefs. If we were to set $p_i^d(\ell)$ to equal a much softer prior over latent labels, $p_i(\ell)$, the minimum would be attained when the two distributions $p_i(\ell)$ and $q_i(\ell)$ are equal: when the prior belief is soft, the trained model $q$ will also be highly uncertain. Turning soft labels into hard training targets, ($p_i^d(\ell) = \mathbb{1}[\ell = \arg\max_\ell p_i(\ell)]$, or by sampling), introduces the opposite bias. Now, the cost would indeed be minimized if the predictions had zero entropy, but learning such a prediction function faces difficulty with overconfident labels which are often wrong, as well as the possibility that certain labels often receive substantial weight in the prior, but never the maximum. These issues are illustrated in Figure D.3.

**Generative modeling resolves the prior's uncertainty.** The approach to classification problems through *generative* modeling, instead of targeting the conditional probability of latents given the data features, assumes that there is a forward (generative) distribution $p(x_i|\ell)$ and optimizes the log-likelihood of the observed features, $\sum_i \log(x_i) = \sum_i \log \sum_\ell p(x_i|\ell) p_i(\ell)$, with respect to the parameters of the forward distribution. The posterior under the model $p(\ell|x_i) \propto p(x_i|\ell) p_i(\ell)$ is then used to infer latent labels for individual data points [44]. As a result, the generative modeling approach does not suffer from uncertainty in the posterior distribution over latents given the input

features, even when the priors $p_i(\ell)$ are soft.[1] Further discussion relating our method with existing generative modeling approaches is given in §C.

However, expressive generative models are typically harder and more expensive to train compared to supervised training of neural networks, as they often require sampling (e.g., sampling of the posterior in VAEs [21] and sampling of the generative model in GANs [8]). Furthermore, the modeling often requires doubling of parameters to express both the forward (generative) model *and* the reverse (posterior) model. And, in case of GANs, the learning algorithms may not even cover all modes in the data, which would prevent joint inference for *all* data points.

## 2.1 OPTIMIZATION UNDER IMPLICIT GENERATIVE MODELS

Suppose that there is a generative model $p(x|\ell)$ of observed features conditioned on latent labels. Optimization of the log-likelihood of observed features, $\sum_i \log(x_i) = \sum_i \log \sum_\ell p(x_i|\ell)p_i(\ell)$, can be achieved by introducing a variational distribution $q_i(\ell)$ over the latent variable for each instance $x_i$, then minimizing the variational free energy, which we review next.

Recall that the free energy, also known as the negative evidence lower bound (ELBO), is defined as

$$F := - \sum_i q_i(\ell) \log \frac{p(x_i|\ell)p_i(\ell)}{q_i(\ell)} = \sum_i \left( \text{KL}(q_i \| r_i) - \log p(x_i) \right), \tag{1}$$

where $r_i(\ell) = \frac{p(x_i|\ell)p_i(\ell)}{\sum_\ell p(x_i|\ell)p_i(\ell)}$ is the posterior distribution under the generative model. As the KL distance is always positive, the free energy is minimized when $q_i = r_i$, in which case the free energy equals the negative log-likelihood of the data.

Free energy minimization is used in various approaches to latent variable modeling. The **expectation-maximization (EM)** [6] algorithm minimizes (1) by iteratively setting $q_i$ to equal the posteriors $r_i$ and updating the parameters of the forward distributions $p(x_i|\ell)$ while keeping $q_i$ fixed, until guaranteed convergence to a local minimum. In applications of EM predating deep generative models, the $q_i$ distributions are not given by a predictor taking $x_i$ as input; they are are auxiliary distributions used only as part of the learning procedure and replaced at each EM iteration. Their dependence on $x_i$ is *implicit* in energy minimization.

In **variational auto-encoders (VAEs)** [21], both the generative model $p(x|h)$ and the posterior $q(h|x)$ are represented as neural networks. Here $p$ is a deep nonlinear generative model with hidden latents $h$ (not necessearily corresponding to labels), and evaluation of the full posterior is intractable. Instead, $q$ is learned as a neural network $q(h|x_i, \theta)$ jointly with $p$. VAE training involves sampling from $q_i(h) = q(x|x_i, \theta)$ in learning to improve the agreement between the forward (generative) model $p$ (and its true posterior $r_i(h)$) and the reverse (posterior) neural network model $q$.

**Implicit generative models.** To derive our training objectives, we will assume the existence of the generative model $p(x|\ell)$ without ever parametrizing it or sampling from it. Instead, we parametrize only the reverse (variational posterior) model $q_i(\ell) = q(\ell|x_i; \theta)$ in the form of a neural net taking $x_i$ as input. The trained model $q$ is the direct output of our algorithms.

Given the variational distributions $q_i$, we can minimize (1) with respect to the forward probabilities $p(x_i|\ell)$ for all $x_i$ and $\ell$ such that $\sum_i p(x_i|\ell) = 1$ for all $\ell$. The optimum is achieved by:

$$p(x_i|\ell) = \frac{q_i(\ell)}{\sum_j q_j(\ell)} . \tag{2}$$

We refer to this as the **implicit generative model** associated to $q$. Much like the posterior is implicit in the free energy minimization in the EM algorithm, here instead the forward model is just a matrix of numbers $p_{i,\ell}$, implicit to free energy minimization – the opposite, in some sense, to what is done in EM optimization. The link $x - \ell$ is left entirely to the neural network $q$ to capture explicitly. The probability mass of the implicit generative model is not uniformly distributed, rather, the data points for which the variational posterior $q_i(\ell)$ is more certain are considered more likely under that latent $\ell$, unless it corresponds to a popular class $\ell$ to which many other data points are also assigned.

---

[1]For intuition, consider a model where $\ell$ is the mixture index in a mixture of high-dimensional Gaussians. If the observations $x_i$ naturally cluster into several well-defined clusters, then the prior $p_i(\ell)$ may be flat, but the posterior distributions under the model assign data points to clusters with high certainty.

The posterior under the implied generative model is

$$r_i(\ell) \propto \frac{p_i(\ell)q_i(\ell)}{\sum_j q_j(\ell)} . \tag{3}$$

Note that we can compute $r_i(\ell)$ by multiplying the re-normalized model outputs with the prior at each instance, so that for each instance $i$ we have two outputs: $q_i$ and $r_i$.

We propose two methods of optimizing the free energy with respect to the parameters $\theta$ by gradient steps. Each method iterates the following:

(1) Calculate the posterior distributions $r_i$ in terms of $q_i$ as in (3)

(2) Update the parameters of $q$ with a gradient step:
   - Option **QR**: $\theta \leftarrow \theta - \eta\nabla_\theta \sum_i \mathrm{KL}(q_i\|r_i)$.
   - Option **RQ**: $\theta \leftarrow \theta - \eta\nabla_\theta \sum_i \mathrm{KL}(r_i\|q_i)$.

```python
# log_q : ( batch_size, n_classes ) log-likelihoods from model
# prior : ( batch_size, n_classes ) prior likelihoods

def ce_loss(log_q, prior):
    return -(log_q * prior).sum(1)

def qr_loss(log_q, prior):
    log_r = (log_q.log_softmax(0) + prior.log()).log_softmax(1)
    return (log_q * log_q.exp()).sum(1) - (log_r * log_q.exp()).sum(1)

def rq_loss(log_q, prior):
    log_r = (log_q.log_softmax(0) + prior.log()).log_softmax(1)
    return (log_r * log_r.exp()).sum(1) - (log_q * log_r.exp()).sum(1)
```

Figure 2: Cross-entropy and implicit **QR / RQ** losses in PyTorch. Note that normalization in (2) is done within a batch, rather than across the entire dataset. In practice, this may be sufficient if batches are large and representative of the diversity in the data. Otherwise, the denominator in (2) may need to be updated in an online fashion.

Gradients of the model parameters are propagated to the expression of $r_i$ through $q_i$ (see Fig. 2). Both losses have a stable point when $q_i = r_i$. A discussion of the relative benefits and limitations of the QR and RQ loss formulations is given in §A, along with practical considerations for implementing these methods.

Option **QR** uses the KL distance in the direction it appears in (1) and thus guarantees continual improvements in free energy and convergence to a local minimum (with the exception for the effects of stochasticity in minibatch sampling). Substituting $r_i$ from (3), the free energy (1) becomes:

$$F = \sum_{i,\ell} q_i(\ell) \log\left(\sum_j q_j(\ell)\right) - \sum_{i,\ell} q_i(\ell) \log\left(p_i(\ell)\right) \tag{4}$$

This criterion does not encourage entropy of individual $q_i$ distributions, but of their *average*. The second term alone would be minimized if the $q$ network could put all the mass on $\arg\max_\ell p_i(\ell)$ for each data point, but the first term promotes diversity in assignment of latents (labels) $\ell$ across the entire dataset. Thus a network $q$ can optimize (4) if it makes different confident predictions for different data points. To illustrate this, consider the case when $p_i(\ell) = p(\ell)$, i.e. all data points have the *same* prior. Then (4) is minimized when $\frac{1}{N}\sum_i q_i(h) = p(\ell)$, and this can be achieved when the network learns a constant distribution $q(\ell|x_i;\theta) = p(\ell)$. But the free energy is also minimized if the network predicts only a single label for each data point with high certainty, but it varies in predictions so that the counts of label predictions match the prior.

As demonstrated in Fig. 1 and in our experiments, avoiding degenerate solutions is not hard. We attribute this to two factors. First, the situations of interest typically involve uncertain, but varying distributions $p_i(\ell)$ which break symmetries that could lead to ignoring the data features $x_i$. Second, the neural networks used as posterior models $q$ come with their own constraints in parametrization (e.g. a multi-layer convolutional net with certain number of filters in each layer), and, the inductive biases as a consequence of gradient descent learning in nonlinear, multilayer networks. In fact, as discussed in §3 and §E.1, even unsupervised clustering is possible with suitably chosen priors that break symmetry, allowing this approach to be used for self-supervised training. See also §C for more on relationships with other approaches.

## 3 SOURCES OF LABEL PRIORS

In this section, we describe a range of machine learning settings where priors $p_i(\ell)$ emerge. These settings are illustrated in experiments in §4, with additional experiments in §E.

**Negative or partial labels (§4.1).** When we are given a set of equally possible labels $L_i$ for each point data point $i$, instead of a single label $\ell_i$, then we set the prior $p_i(\ell) = \frac{1}{|L_i|}\mathbb{1}[\ell \in L_i]$. An extreme example is when one negative label is given, as shown in Fig. 1.

**Joint labels and learning from rankings (§4.2).** Priors may also come in the form of joint distributions over labels of multiple instances. For example, ranking supervision – the knowledge of which example in a pair is greater with respect to an ordering of the labels – gives prior beliefs about *pairs* of labels. Suppose our data is organized into pairs of images of digits $T_j = \{x_{j,1}, x_{j,2}\}$, and for each pair we are told which image represents the digit (0–9) which is greater. This sets a prior $p(\ell_1, \ell_2)$ over pairs of labels in each pair, represented by either an upper or a lower triangular matrix, depending on which digit in the pair is known to be greater, with all nonzero entries equal to $1/55$.

We assume the implicit generative model has the form $p(x_1, x_2|\ell_1, \ell_2) = p(x_1|\ell_1)p(x_2|\ell_2)$. We aim to fit a posterior model $q(\ell|x; \theta)$. For each pair $T_j$, we have two outputs of the predictor network, $q(\ell_1|x_{j,1})$ and $q(\ell_2|x_{j,2})$, for the two images in the pair. The joint posterior under the implicit generative model is

$$r_j(\ell_1, \ell_2) \propto p(\ell_1, \ell_2)p(x_{j,1}|\ell_1)p(x_{j,2}|\ell_2) \propto \frac{p(\ell_1, \ell_2)q(\ell_1|x_{j,1})q(\ell_2|x_{j,2})}{\sum_j q(\ell_1|x_{j,1}) \sum_j q(\ell_2|x_{j,2})}, \tag{5}$$

and we can now use **QR** or **RQ** loss to fit $q(\ell_1|x_{j,1})$ to the marginal $r_j(\ell_1)$ and $q(\ell_2|x_{j,2})$ to $r_j(\ell_2)$.

**Coarse data in weakly supervised segmentation (§4.3, §E.2, §E.4).** We often have side information $z$ associated to each instance $i$ that allows setting the priors $p_i(\ell) = p(\ell|z_i)$ for each point directly by hand. These include situations when we have beliefs about labels for different points, as in the *Seducer* example (Fig. 1), but also more interesting weak supervision settings, such as ones that sometimes arise in remote sensing (§4.3) and medical pathology (§E.2) applications. For example, in a task of segmenting aerial imagery into land cover classes, we often have coarse labels $c$ associated to large *blocks* of pixels, but not the target labels $\ell$ for individual pixels. If the conditional $p(\ell|c)$ is known, it sets a belief about the high-resolution labels $\ell$ for pixels in a block of class $c$.

**Fusing models and data sources (§4.4, S4.5).** Auxiliary information $z$ may not always come with a known correspondence $p(\ell|z)$. In the land cover mapping problem, auxiliary information may include different modalities and resolutions (road maps, sparse point labels, etc.). While these sources can be fused into a prior by hand-coded rules, the prior may be more accurately set as the output of a model $p(\ell|z_i)$ *trained* on a separate dataset of points $(\ell_i, z_i)$. This is especially useful when the data $x_i$ (e.g. satellite imagery) is more informative about the latents $\ell_i$, but is prone to domain shift problems, while the auxiliary data $z_i$ does not suffer from domain shift issues but is not sufficient on its own to predict the labels. In a text classification problem, $z_i$ might be the encoding of text $x_i$ by a pretrained language model, and $p(\ell|z_i)$ a noisy distribution over labels given by their likelihoods under the language model as continuations of a prompt.

**Priors for self-supervision (§E.1).** In §2.1 we discussed the pitfalls of using a constant prior $p_i(\ell) = p(\ell)$ for all data points in training models under the **QR** loss as a potential method for unsupervised clustering. However, in §E.1 we give an example of *joint* learning of the posterior model $q$ and an energy model (Markov random field) on the latent labels $\ell_i$ that expresses local structure of labels in an image. This results in unsupervised clusterings that are useful in downstream segmentation tasks. Such an approach is an example of a benefit of generative modeling – the possibility of learning of a parametrized distribution over latents – being inherited by implicit generative models.

**Priors with latent structure (§E.3).** Implicit generative modeling allows building hierarchical latent structure into the prior (another benefit of classical generative models), as we demonstrate in §E.3 on a video segmentation task. The prior is an admixture of possible segmentations with a structure similar to [17], but using a set of mask proposals $p(\ell_i|m)$ from a Mask R-CNN model [11], indexed by a latent $m$. The prior is $p_i(\ell) = \sum_m p(\ell_i|m)p(m)$, where $p(m)$, a probabilistic selection of the masks for the admixture in the given frame, is estimated by minimizing the free energy.

## 4 EXPERIMENTS

### 4.1 PARTIAL LABELS IN MNIST AND CIFAR-10

In this illustrative experiment, we compare algorithms for learning with partial labels on two 10-way image classification datasets, MNIST and CIFAR-10. To each training example $x_i$, we randomly assign a set $N_i$ of $k$ negative labels, chosen from the 9 labels distinct from the ground truth. The prior $p_i(\ell)$ is set to be uniform over $\ell \notin N_i$ and 0 for $\ell \in N_i$. We vary $k$ from 1 (one negative label per example) to 9 (one-hot prior, full supervision). The data of $k$ negative labels carries $-\log_2(1 - k/10)$ bits of label information – if $k = 1$, 22× less label information than in the fully supervised setting.

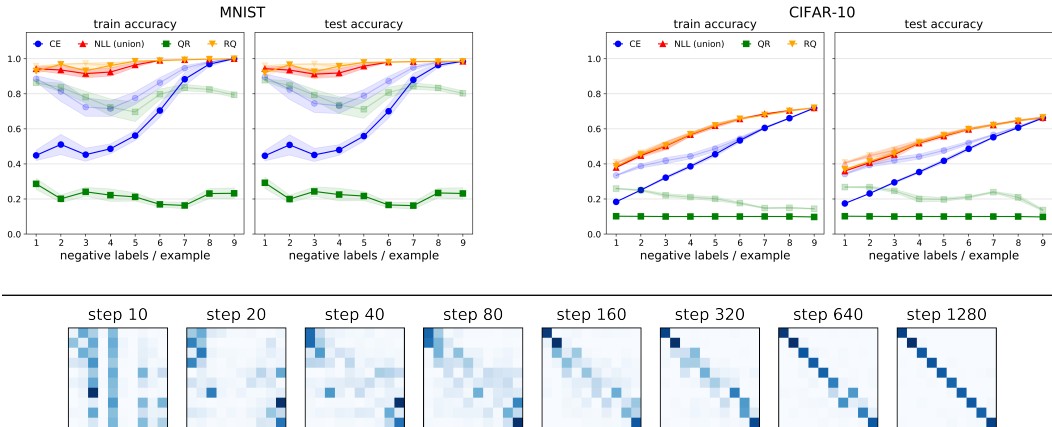

Figure 3: **Above:** Accuracies of MNIST and CIFAR-10 classifiers trained with varying numbers of negative labels per example; the lighter variant of each color and marker shows the peak accuracy over 300 training epochs. (Average of 10 runs with standard error region.) **Below:** Confusion matrices of MNIST classifiers in the course of training on batches of 128 pairs of digits. The trajectory of convergence to the diagonal shows that uncertainty is first resolved for the digits 0 and 9, then 1 and 8, etc.

For both datasets, the base model $q$ is taken to be a small convolutional network, with four layers of ReLU-activated $3 \times 3$ convolutions with stride 2 and a linear map to the 10 output logits (~33k learnable parameters for MNIST, ~34k for CIFAR-10). We experiment with four training losses:
- **CE:** cross-entropy between predictions $q(\ell|x_i; \theta)$ and the prior $p_i(\ell)$.
- **NLL (union):** negative logarithm of the sum of likelihoods assigned by $q$ to labels in $\ell \notin N_i$, or, equivalently, $\log \sum_\ell p_i(\ell) q(\ell|x_i; \theta)$, as done, e.g., by [15; 19].
- The **QR** and **RQ** losses.

The **CE**, **NLL (union)**, and **RQ** losses are equivalent when $k = 9$, and the **RQ** and **NLL (union)** losses are equivalent when $\sum_i q_i(\ell)$ is uniform over $\ell$, which approximately holds late in training.

All models are trained for 300 epochs on batches of 256 images with the Adam optimizer [20] and a learning rate of $10^{-4}$. After each epoch, we compute the accuracy of the predictor $q$ on the ground truth labels in the train and test sets. Fig. 3 shows the final accuracies, as well as the maximum accuracies over epochs, averaged over 10 choices of partial label sets and random initializations. Models trained on **RQ** loss perform best, with the greatest benefit over **CE** seen for very few negative labels. We hypothesize that the small advantage of **RQ** over **NLL (union)** loss can be attributed to regularization in early training. Meanwhile, **QR** performs as well as **CE** for very uncertain priors at the peak epoch, but its predictions degenerate with longer training.

## 4.2 Multiple-instance supervision: Learning from ranks

We train a CNN of the same architecture as in §4.1 on MNIST, but with the only supervision coming in the form of pairs of images in which it is known which image represents the greater digit. The training set of 60k images is divided into pairs that are fixed throughout the training procedure; each digit appears in exactly one pair. We optimize to match the predictor $q$ with the implicit generative model's posterior (5) using the **RQ** loss. Fig. 3 shows the confusion matrices at initial iterations of training. The learned classifier has 97% accuracy on both training and testing sets, which means that from pairwise comparisons alone, we can group the digit images and place them in order.

## 4.3 Label super-resolution: Chesapeake Bay land cover mapping

We benchmark our method's performance on the Chesapeake Land Cover dataset[2], a large 1m-resolution land cover dataset used previously for label super-resolution [27; 42]. It con-

[2]https://lila.science/datasets/chesapeakelandcover

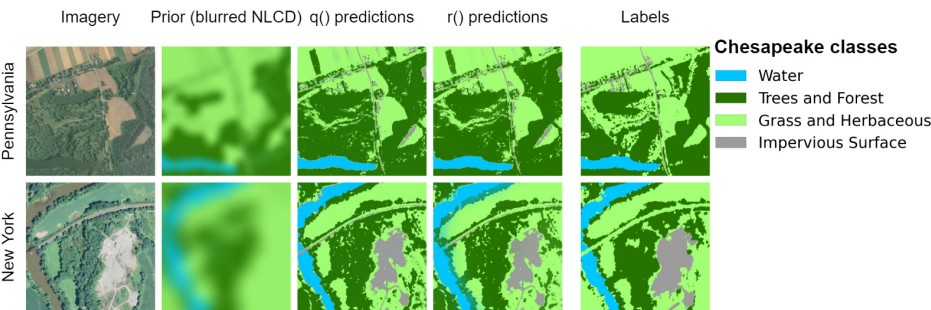

Figure 4: Predictions of models trained with **QR** loss on the NLCD-only prior in the Chesapeake region, shown on regions of 1000×1000 pixels in Pennsylvania and 500×500 pixels in New York.

sists of several aligned data layers, including: NAIP (4-channel high-resolution aerial imagery at about 1m/px), NLCD (16-class, 30m-resolution coarse land cover labels), and high-resolution land cover labels (LC) in four classes. The task is to train high-resolution segmentation models, in the four target classes, using only NLCD labels as supervision. Although the NLCD layer is at 30× lower resolution than the imagery and target labels and follows a different class scheme, the cooccurrence statistics of NLCD classes $c$ and LC labels $\ell$ are assumed to be known.

To form a prior over land cover classes $\ell$ at each pixel position, we map the NLCD classes to probabilities over the target LC classes using these known cooccurrence counts and apply a spatial blur to reduce low-resolution block artifacts (Fig. 4, "Prior"). We then train small convolutional networks (receptive field $11 \times 11$) to predict high-resolution land cover from input imagery. We evaluate both the **QR** and **RQ** variants of our approach on the two states that comprise the "Chesapeake North" test set: Pennsylvania (PA) and New York (NY), and the two states combined, after picking hyperparameters based on an independent validation set in Delaware (details in Appendix D.1.3). A depiction of the data and prediction results is given in Figure 4.

Table 1 compares our performance against the algorithmic technique with the best published performance on the Chesapake dataset, self-epitomic LSR [28] and the hard naïve baseline from [27]. Self-epitomic LSR, a generative modeling approach that explicitly produces likelihoods $p(x|\ell)$, analyzes small patches of data by making a large number of comparisons between sampled $7 \times 7$ image patches and *all other* image patches. It does not produce a trained feedforward inference model, and the inference procedure is at least an order of magnitude slower than evaluation of our convolutional model. The hard naïve baseline maps the NLCD classes to LC classes based on a given concurrence matrix, then trains a standard semantic segmentation model on these pseudo-labels.

Not only does training on the **QR** loss achieve comparable performance with self-epitomic LSR (Table 1), but the implicit generative model for $p(x|c)$ from (2) is largely consistent with the epitomic generative model (Fig. D.4). Moreover, our method handles *batched input*, where self-epitomic LSR trains on one tile at a time, and similar approaches have been shown to degrade in performance and exhaust computation capacity when training on multiple tiles [28]). Thus, optimization under an implied generative model has the computational advantage of scaling naturally to large training data while maintaining the benefits of leading generative modeling approaches. (See also §E.2.)

## 4.4 DATA FUSION AND LEARNED PRIORS: ENVIROATLAS

In this set of experiments, we augment NLCD with information about the presence of buildings, road networks, water bodies, and waterways from public sources (see Fig. 5 and §D.1.1). To evaluate the ability of models to generalize to new geographic regions, we use 1m 5-class land cover labels from the geographically diverse EnviroAtlas dataset [39] in four cities in the US: Pittsburgh, PA, Durham, NC, Austin, TX, and Phoenix, AZ. The NLCD-based prior model from §4.3 is augmented with the auxiliary information to obtain a hand-coded prior for each image (see §D.1.2.) These types of priors can be made everywhere in the United States, while hard labels are rarely available.

The standard alternative to performing local inference under such priors is to simply apply supervised models trained on hard labels elsewhere, hoping that the domain shift is tolerable. Table 2

Table 1: Pixel accuracy and class mean intersection over union on the Chesapeake Land Cover dataset. All models use only coarse NLCD labels as supervision. For our proposed methods, we evaluate both the trained predictor ($q_i$) and the posterior under the implicit generative model ($r_i$).

| Model | PA acc % | PA IoU % | NY acc % | NY IoU % | Chesapeake acc % | Chesapeake IoU % |
|---|---|---|---|---|---|---|
| Self-epitomic LSR [28] | 86.2 | 67.6 | 86.4 | 70.5 | 86.3 | 69.7 |
| Hard naïve [27] | 85.3 | 63.0 | 83.6 | 59.8 | 83.6 | 59.7 |
| **QR** ($q$) | 84.2 | 66.6 | 86.2 | 71.0 | 84.6 | 67.7 |
| **QR** ($r$) | 85.0 | 68.1 | 86.9 | 72.4 | 85.4 | 69.4 |
| **RQ** ($q$) | 82.2 | 63.4 | 79.0 | 61.4 | 78.7 | 59.4 |
| **RQ** ($r$) | 82.3 | 63.5 | 79.0 | 61.6 | 78.8 | 59.7 |

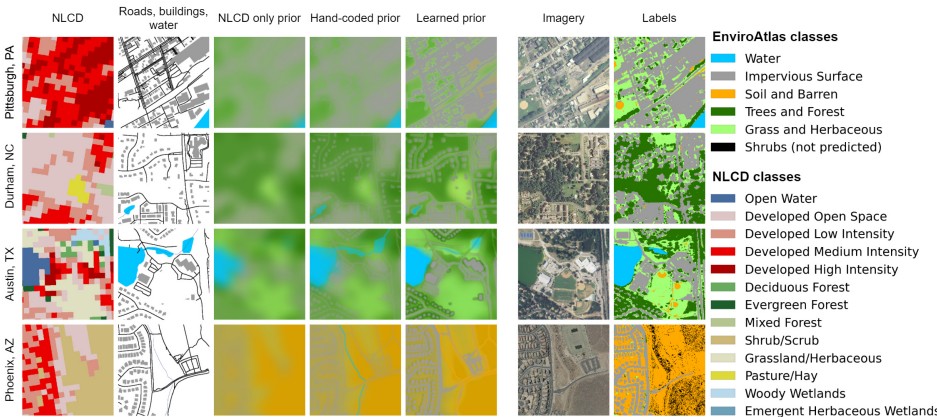

Figure 5: Prior generation process for land cover mapping: "NLCD only prior" as used in §4.3 and "hand-coded prior" and 'learned prior' as used in §4.4.

compares the performance of a model (of the same architecture as in §4.3) trained on Pittsburgh high-resolution data (HR) in each of the three other cities with that of models tuned on the hand-coded prior in each other city. The **QR** method trained on the local handmade prior outperforms the HR model in each evaluation city. This may be attributed to the extra data in each city given to our method in the form of prior beliefs. To isolate this effect, we also compare to a high-resolution model that consumes the prior belief to *input* data, concatenated with the NAIP imagery (HR + aux). While the HR + aux model does increase performance substantially from the HR model with NAIP imagery alone as input, the **QR** model remains the highest-fidelity approach. These results illustrate that information that generalizes across domains may find its best use within a separate model – to build a prior in our setting – and then used to supervise local inference.

A prior belief could be crafted by a domain expert to reflect the uniquities in geographic and structural features for each city. We emulate incorporating such context-specific knowledge by training (on a disjoint set of instances) a neural network that consumes the inputs to the handmade prior function (NLCD and auxiliary map data), and predicts high-resolution labels (Figure 5, "Learned prior"). Alongside structural interactions between the inputs that generalize across cities (e.g. tree canopy supersedes rivers, road supersede water), the learned prior captures region-specific knowledge (e.g., buildings in Durham, tend to be have grass surrounding them and trees farther out, while in Austin, this is reversed, and in Phoenix, riverbeds surrounded by barren land are likely to be dry). As shown in Table 2, these tailored prior beliefs tend to increase scores.

The final row in Table 2 benchmarks the performance of a high-resolution land cover model trained on imagery and labels over the entire contiguous US [42]. This large model takes NAIP, Landsat 8 satellite imagery, and building footprints as inputs. Small, local models with priors created from only weak supervision outperform the US-wide model in all cities. (See §D.1.4 for evaluation details.)

Table 2: Land cover classification experiments for generalizing across cities. In each column, the score of the best model not depending on auxiliary data as input is *italicized*. The score of the best overall model is **bolded**. (A larger set of experimental results is given in Table D.1.)

| Train region | Model | Durham, NC | | Austin, TX | | Phoenix, AZ | |
|---|---|---|---|---|---|---|---|
| | | acc % | IoU % | acc % | IoU % | acc % | IoU % |
| Pittsburgh | HR | 72.9 | 34.8 | 71.9 | 36.8 | 6.7 | 7.5 |
| (supervised) | HR + aux | 78.9 | 47.7 | 77.2 | 50.6 | 66.0 | 27.9 |
| Local | **QR** $(q)$ | *78.9* | 46.6 | 79.3 | *51.7* | 73.2 | 39.1 |
| (hand-coded prior) | **QR** $(r)$ | **79.1** | 48.8 | 79.6 | **53.0** | 73.4 | 40.6 |
| Local | **QR** $(q)$ | 78.7 | *47.4* | *79.8* | 51.1 | *74.6* | *39.6* |
| (learned prior) | **QR** $(r)$ | 78.9 | **49.6** | **80.3** | 52.6 | **75.3** | **42.2** |
| Full US [42] | U-Net Large | 77.4 | 48.8 | 76.4 | 51.4 | 24.5 | 23.2 |

Table 3: F1-scores of various models on the coarsely supervised text classification task. The first five columns are taken from [32]. The last two columns use the GPT-2 prior defined in §4.5 as weak supervision with cross-entropy and **RQ** loss, respectively (mean of 10 random initializations).

| | pseudolabeling | | | | | GPT-2 prior, trigram features | | |
|---|---|---|---|---|---|---|---|---|
| | WeSTClass [33] | ConWea [31] | LOTClass [34] | X-Class [51] | C2F [32] | prior argmax | CE | **RQ** |
| Micro-F1 % | 76.23 | 73.96 | 15.00 | 91.16 | 92.62 | 86.33 | 87.18 | **93.18** |
| Macro-F1 % | 69.82 | 65.03 | 20.21 | 81.09 | **87.01** | 77.61 | 77.90 | 84.26 |

## 4.5 UNCERTAIN TEACHERS: TEXT CLASSIFICATION

This experiment follows the very recent work of [32] and illustrates the effectiveness of learning on prior beliefs beyond computer vision. We work with a dataset of ~12k New York Times news articles. Each article belongs to one of 20 fine categories (e.g., 'energy companies', 'tennis','golf'), which are grouped into 5 coarse categories (e.g., 'business', 'sports'). The goal is to train text classifiers that predict fine labels, but only the coarse label for each article is available in training.

Some external knowledge about the fine categories is necessary to resolve the coarse labels into fine labels. Past work on this problem [33; 31; 34; 51] has trained supervised models on pseudolabels created by mechanisms such as propagation of seed words and querying large pretrained language models. On the other hand, [32] create training data by sampling additional *features* (articles) from a finetuned version of the large generative language model GPT-2 [41] conditioned on fine categories, then tune a classifier based on the almost equally large model BERT [7] in a supervised manner.

We obtain comparable results with an elementary predictor, far less computation, and no finetuning of massive language models. We form a prior $p_i(\ell)$ on the fine class $\ell$ of each article $x_i$ by querying GPT-2 for the likelihood of each fine category name $\ell$ compatible with the known coarse label following the prompt "[article text] Topic: " and normalizing over $\ell$. We then divide $p_i(\ell)$ by the mean likelihood of $\ell$ over all articles $x_i$ and renormalize. We represent each article as a vector of alphabetic trigram counts ($26^3$ features, of which only 8k are ever nonzero) and train a logistic regression with the **RQ** objective against the 'GPT-2 prior'. After ten epochs of training (~10s on our hardware), the trained classifier nears or exceeds the performance of models requiring at least $100\times$ longer to train, not to mention to generate pseudo-training data for (Table 3).

## 5 CONCLUSIONS

We found that the generative distribution in a free energy criterion can be left implicit to the minimization process in posterior (discriminative) model training. This allowed us to unite the training of neural networks $q(\ell|x_i; \theta)$ for prediction of labels $\ell$ from features $x$ and the modeling of the prior $p_i(\ell)$, possibly with its own latent structure. Implicit modeling of the conditional generative distributions removes the burden of training accurate (and therefore large or deep) generative models, but allows natural generative approaches to modeling priors. We expect interesting new modeling paradigms that integrate different modalities and models to arise from this formulation.

REPRODUCIBILITY STATEMENT

This submission comes with two code directories that illustrate our algorithms for partial-label learning and weakly supervised segmentation. The provided code is sufficient to reproduce results resembling those in Fig. 1. Upon publication, we intend to release code for all land cover mapping experiments that is compatible with the TorchGeo library.

ETHICS STATEMENT

The authors anticipate that the immediate consequences arising from this work will have nonnegative social impacts. In general, learning from weak, coarse, or imprecise data may be susceptible to data biases in differential quality or precision in data sources across sub-populations or geographies. We encourage further work to examine these issues, especially in cases where uncertain labels are used in machine learning systems that impact people or decision-making systems. Overall, we hope that the ability to learn from a variety of weak data sources can aid applicability of machine learning systems in diverse settings.

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

## A  PRACTICAL CONSIDERATIONS

**Mini-batches:**   Figure 2 shows a PyTorch implementation of the QR and RQ loss functions, where loss is computed over *batches* of training data. Our experiments validate that so long as these batches are large enough to include enough diversity of $(x_i, p_i(l))$ pairs, our method works when eq. (2) and eq. (3) are applied directly to batches. As discussed in §4.4, handling batched input is important for leveraging the scale of large training datasets. As discussed in the caption of Figure 2, should mini-batch training become an issue in future implementations, it may be beneficial to estimate the denominator of eq. (2) across multiple batches.

**Relative benefits/limitations of the QR and RQ loss formulations:**   The algorithm presented in §2.1 details two loss options: a **QR** option and an **RQ** option, both with unique strengths. The QR algorithm is guaranteed to converge as each step reduces loss (except for randomness in the learning algorithm). The RQ algorithm, on the other hand, has the appealing property that it reduces to standard minimization of cross entropy loss in the case of hard labels. In §C, we discuss connections between QR option and variational auto-encoders (VAEs), and between the RQ option and the wake-sleep algorithm. Ultimately, though, we find that which option works better may depend on the application, with RQ working across all applications we tried but sometimes being slightly beaten by QR.

**Simple ways to avoid degenerate solutions**   As discussed in §2.1, minimizing eq. (1) can lead to degenerate solutions. However, avoiding these solutions can be quite simple, and in most of our experiments we did not make any interventions to explicitly avoid such local minima. In a targeted experiment in table D.1 we show that pre-training on hard labels (even out-of-domain) or using sharper learned priors can help break symmetries during early training phases. When hard labels are not available, one could similarly start the training process with a cross-entropy loss on the prior belief, and then switch to RQ or QR loss. The intuition is that first training to minimize cross-entropy breaks the symmetry at the start, while implicit generative modeling sharpens the predictions in later iterations.

## B  ADDITIONAL RELATED WORK

There are several approaches to learning with uncertain, weak, or coarse labels under different assumptions and settings. Work on partial-label learning often employs loss functions that aim to decrease prediction entropy [35; 54; 55]; In contrast to our work, these approaches do not use a generative formulation in these loss functions, makes them ill-suited to problems with more varied forms of uncertainty encoded in priors. Another line of work studies the generation and use of pseudo labels in learning settings. Specifically, [59] relies on a domain-specific augmentation procedure for semantic segmentation with image-level labels, and, [56] studies unsupervised clustering applied to object re-identification. Application-specific solutions also include object detection in remote sensing images [10] and change detection with temporal satellite imagery [57; 1; 22].

In our experimental setups, we chose a mix of baselines to compare algorithm design and benchmark performance on certain tasks.

To compare our approach on an *algorithmic basis*, we compare to the negative logarithm of the sum of likelihoods (NLL), which is used in prior works to handle multiple ambiguous labels [15] and negative labels [19]. As we discuss in §4.1, NLL and RQ are equivalent when $\sum_i q_i(\ell)$ is uniform over $\ell$, evidenced by the comparable performance between the two in Figure 3. We compare to self-epitomic LSR [28] as an algorithmic comparison by which to contrast our method with an "explicit" generative modeling approach. Our similar performance to self-epitomic LSR in regimes where self-epitomic LSR has been shown to perform well (super-resolution in land cover mapping (§4.3) and the tumor-infiltrating lymphocytes task (§E.2)) is an important validation of our motivation in §2.

To benchmark *performance* of our approach across tasks, we compare to state-of-the-art pseudo-labeling methods in supervised text classification  (see §4.5), an established 1m resolution map of land cover predictions across the United States [42] and best-performing published results for the land cover mapping tasks we study [28; 43], the best known published results for the tumor-

infiltrating lymphocyte segmentation task ([27; 28]), and a host of comparisons for the video instance segmentation task (see Table E.3 for a full list).

Lastly, it is worth noting that the term "implicit generative model" has also been used in prior literature to refer to amortized sampling procedures for nonparametric (or not specified) energy functions, such as generative adversarial models (e.g., [45]). Although we do not make an explicit connection with such models, our formulation also does not assume a parametrization of the data distribution. However, we assume tractability of sampling from a posterior over certain distinguished latents (classes) conditioned on observed data (features, e.g., images), rather than directly sampling latents.

## C  RELATIONSHIPS WITH EM, VAEs, AND WAKE-SLEEP ALGORITHM

As discussed in §2.1, the **QR** loss guarantees continual improvements in the free energy (1). On the other hand, option **RQ** is equivalent to performing a gradient step on the cross-entropy of $q_i$ and $r_i$ and a gradient step on the *negative* entropy of $r_i$. In the case that the priors $p_i(\ell)$ are hard (supported only on one ground truth label), the same is true of $r_i$, and the **RQ** loss is equivalent to cross-entropy. This option reverses the KL distance in a manner reminiscent of the training procedure in the wake-sleep algorithm [12], where parameter updates for the forward and reverse models are iterated, but the KL distance optimized always places the probabilities under the model being optimized in the second position in the KL distance (inside the logarithm), so that the generative and the inference models each optimize log-likelihoods of their predictions. The wake-sleep algorithm, however, also trains a generative model rather than treating it as an auxiliary distribution like we do, and that requires sampling. As opposed to VAEs, the wake-sleep algorithm samples the generative model, not the posterior.

It is interesting to contrast our approach to the EM formulation. In standard EM, the $q$ distributions are considered auxiliary, rather than parametrized as direct functions of the inputs $x$. The $q_i(\ell)$ is simply a matrix of numbers normalized across $\ell$. Its dependence on the data $x$ is only implicit in the iterative re-estimation of the minimum of the free energy. The link $x - \ell$ is modeled explicitly in the parametrized forward distribution $p(x|\ell)$. We instead treat forward probabilities $p(x_i|\ell)$ as auxiliary parameters, a matrix of numbers $p_{i,\ell}$ normalized across $i$ that we fit to minimize the free energy at each data point, and optimize only the parameters of the $q$ model which explicitly models the link $x - \ell$. This then allows us to capture nonlinear (and 'deep') structure and benefit from inductive biases inherent to training deep models with SGD, but without the cost of training an actual parametrized generative model and other problems associated with deep generative model fitting. The resulting $q$ network approximates the posterior in a generative model – which (locally) maximizes the log likelihood of the data – and it is usually highly confident (as seen in Fig. 1).

The implicit modeling of the posterior in EM does not lead to over-fitting of the generative model. But, given that degenerate solutions to optimization with implicit generative models are possible when the prior is constant across all data points (§2.1), we can imagine that our approach of implicit generative modeling might lead to degenerate solutions. As demonstrated in Fig. 1 and in our experiments, avoiding degenerate solutions is not too hard. We attribute this to two factors. First, the situations of interest typically involve uncertain, but varying distributions $p_i(\ell)$ which break symmetries that could lead to ignoring the data features $x_i$ as in the example above of a constant prior. Second, the neural networks to be used as posterior models $q$ come with their own constraints in parametrization, and, equally importantly, the inductive biases that come as a consequence of gradient descent learning in nonlinear, multilayer networks.

## D  EXPERIMENT DETAILS

### D.1  LAND COVER MAPPING

#### D.1.1  DATASETS

**Imagery Data**  Our land cover mapping experiments use imagery from the National Agriculture Imagery Program (NAIP), which is 4-channel aerial imagery at a $\leq$ 1m/px resolution taken in the United States (US).

**Chesapeake Conservancy land cover dataset** The Chesapeake Conservancy land cover dataset consists of several raster layers of both imagery and labels covering parts of 6 states in the Northeastern United States: Maryland, Delaware, Virginia, West Virginia, Pennsylvania, and New York [42][3]. The raster layers include: high resolution (1m/px) NAIP imagery, high resolution (1m/px) land cover labels created semi-autonomously by the Chesapeake Conservancy, low resolution (30m/px) Landsat-8 mosaics imagery, low resolution (30m/px) land cover labels from the National Land Cover Database (NLCD), and building footprint masks from the Microsoft Building Footprint dataset. The dataset is partitioned into train, validation, and test splits per-state, where each split is a set of $\approx$ 7km $\times$ 6km *tiles* containing the aligned raster layers.

**EPA EnviroAtlas data** The EnviroAtlas land cover data consists of high resolution (1m/px) land cover maps over 30 cities in the US, and is collected and hosted by the US Environmental Protection Agency (EPA) [39]. A detailed description of the dataset and its land cover definitions is provided by [40]. As with most high-resolution land cover datasets (including the Chesapeake Conservancy land cover labels), the EnviroAtlas land cover labels are themselves derived by remote sensing and learning procedures, and thus are not themselves a perfect "ground truth" representation of land cover. For example, the estimated accuracy of the provided labels is 86.5% in Pittsburgh, PA, 83.0% in Durham, NC, 86.5% in Austin, TX, and 69.2% in Phoenix, AZ [40].

The high-resolution label files were aligned to match the extent of the NAIP tiles from the closest available years to the years that the EnviroAtlas labels were collected: for Pittsburgh, PA and Phoenix, AZ, we used data from 2010 and for Durham, NC and Austin, TX, we used data from 2012. We chose these four cities to get a wide coverage across the United States (US), and due to a mostly consistent set of classes being used between the four cities.

**National Land Cover Database (NLCD)** The National Land Cover Database is produced by the United States Geological Survey (USGS) and uses 16 land cover classes. Maps are generated every 2-3 years, with spatial resolution of 30m/px. Data and more information can be found at: `https://www.usgs.gov/centers/eros/science/national-land-cover-database`.

**Microsoft Building Footprint dataset** The Microsoft Building Footprint dataset consists of predicted building polygons over the continental US from Bing Maps imagery. As of the time of writing, the most updated Microsoft Building Footprints dataset in the US can be accessed at: `https://github.com/Microsoft/USBuildingFootprints`.

**Open Street Map (OSM) data** Open Street Map (`https://www.openstreetmap.org/`) is an ongoing effort to make publicly available and editable map of the world, generated largely from volunteer efforts. The data is available under the Open Database License. From the many different sources of information provided by OSM [9], we download raster data for road networks, waterways, and water bodies, using the OSMnx python package [2].

**Data splits and data processing** For experiments using the Chesapeake Conservancy dataset (Table 1), we used established train, test, and validation splits. In particular, we used the 20 test tiles in New York (NY) and the 20 test tiles in Pennsylvania on which to conduct our experiments. Here a *tile* matches the extent of a NAIP tile, roughly 7km $\times$ 6km. To facilitate comparison of our results with previous published results on this dataset, we condensed the labels into four classes: (1) water, (2) impervious surfaces (roads, buildings, barren land), (3) grass/field, and (4) tree canopy.

For experiments with the EnviroAtlas dataset (Table 2), we aligned the high resolution land cover data, NLCD, OSM, and Microsoft Building Footprints data with NAIP imagery tiles, matching years as closely as possible to the EnviroAtlas data collection year for NLCD and NAIP. We instantiated a split of 10 train, 8 validation, and 10 test tiles in Pittsburgh, and 10 test tiles in Durham, NC, Austin, TX, and Phoenix, AZ. For Pittsburgh we assigned tiles to splits randomly from the set of 28 tiles that had no missing labels. There were not enough such tiles in Durham to follow the same procedure, so we chose the ten evaluation tiles at random from a set with no number of missing labels per tile. For Austin and Phoenix, we chose the 10 evaluation tiles at random from the tiles in each city that had no agriculture class (as it is not present in Pittsburgh or Durham) and no missing labels.

---

[3]Dataset can be downloaded from: `https://lila.science/datasets/chesapeakelandcover`.

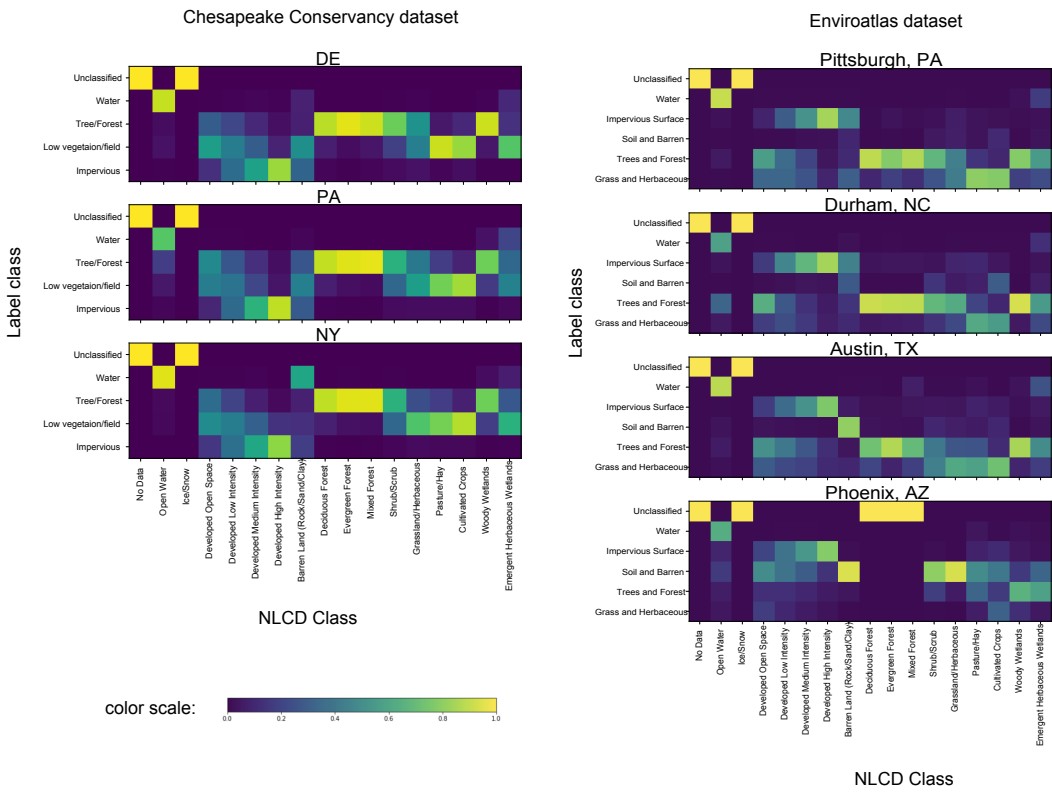

Figure D.1: Cooccurrence matrices between NLCD classes and high resolution land cover labels for each region we study.

We set aside 5 separate tiles in each city for use in "learning the prior" (in Pittsburgh these 5 tiles are a subset of the 8 validation tiles). As above, each tile corresponds to one NAIP tile. The tiles in these constructed sets for Pittsburgh, Durham, and Austin contain five unique labels: (1) water, (2) impervious surfaces (roads, buildings), (2) barren land, (4) grass/field, and (5) trees. Phoenix additionally has "shrub" class; when forming the prior we merge this class with trees, and we ignore the shrub class when evaluating in Phoenix. We cropped all data tiles to ensure no spatial overlap in any tiles between or within the train/val/test splits.

### D.1.2 FORMING THE PRIORS

To form the priors for the land cover classification tasks, we first spatially smooth the NLCD labels by applying a 2D Gaussian filter (with standard deviation 31 pixels) across every channel in a one-hot representation of the NLCD classes. The main reason for applying this smoothing is to reduce artifacts due to the 30m$^2$ boundaries of the NLCD data: to undo the blocking procedure induced by the aggregation to 30m × 30m extents, to incorporate the spatial correlations between nearby NLCD blocks, and to remove erroneous sharp differentials between inputs that can cause artifacts during later training stages.

We then remap the blurred NLCD layers to the classes of interest by multiplying by a matrix of co-occurrence counts between the (unblurred) NLCD data and the high resolution labels in each region. For the Chesapeake region, we use the train tiles provided with the Chesapeake Conservancy land cover dataset to define co-occurrence matrices in NY and PA. For EnviroAtlas, we compute co-occurrences using the entire city (excluding tiles with agriculture in Phoenix AZ, and Austin, TX). The co-occurrence matrices for each region we study are shown in Figure D.1.

The priors for the Chesapeake Conservancy dataset are then generated by normalizing the blurred and remapped NLCD data so that summing over all five classes gives probability 1 for each pixel.

For the EnviroAtlas data, we augment this prior with publicly available data on buildings, road networks, water bodies, and waterways. We obtain building maps from the Microsoft Buildings Footprint database and road, water bodies, and waterways data from Open Street Map, using the OSMnx tool [2] to download the data (see Appendix D.1.1). We apply a small spatial blur to each of these input sources to account for (a) vector representation of roads and waterways being unrealistically thin, and (b) possible data-image misalignment on the order of pixels. Where this results in probability mass on impervious surfaces or water, we add these probability masses to the blurred NLCD prior, and then renormalize to obtain a valid set of probabilities for each pixel.

In §4.4, we describe a method for "learning the prior," which uses a more sophisticated process to aggregate the individually weak and coarse inputs that we use in the handmade prior. In this method, we train a neural net to take as input the blurred, remapped NLCD representation (5 classes) concatenated with the 4 classes of additional data: buildings, roads, waterways, water bodies, and to predict high-resolution labels in each city. We train these networks using 5 tiles of imagery and high-resolution labels from the EnviroAtlas Dataset in each city which are distinct from the 10 test tiles in each city. The training procedure for these prior generation networks is described in in §D.1.3. To create the priors that we then train our method on ('learned prior' rows in Table 2) we ran these learned models forward on (blurred and remapped NLCD, buildings, roads, waterways, and waterbodies) input for each of the 10 evaluation tiles in each city.

### D.1.3 EXPERIMENTAL PROCEDURE

We use priors generated as described in Appendix D.1.2, with Gaussian spatial smoothing standard deviation of 31 pixels, and co-occurrence matrix determined via the training splits in each city/state. We apply an additive smoothing constant of 1e-4 to that is applied pixel-wise the the probability vectors output by the neural network as well as to the prior probability vectors used as the model supervision data. This additive smoothing constant ensures that there are no extremely low probability classes in either the prior or the predicted outputs during training.

Experiments summarized in Table 1 and Table 2 use a 5-layer fully connected network with kernel sizes of 3 at each layer, 128 filters per layer, and leaky ReLUs between layers. Note that the receptive field of this model is only $11 \times 11$ pixels. We use batch sizes of 128 instances during training, where each image instance is a cropped $128 \times 128$ pixels from a larger tile. Training is and model evaluation is done within the torchgeo framework for geo-spatial machine learning [48]. All models use the AdamW optimizer [24] during training and torchgeo defaults unless otherwise noted.

**Comparison to previous label super-resolution for LC mapping** To obtain the parameter setting used for the runs in New York (NY) and Pennsylvania (PA) in Table 1, we first perform a hyperparameter search with the 20 tiles test set in Delaware (DE) from the same overall dataset. We use a learning rate schedule that decreases learning rate when the validation loss plateaus, as well as early stopping to prevent over training of models. Of the grid of learning rates in [1e-3, 1e-4,1e-5], we describe below, we pick learning rate as 1e-4 for both **QR** and **RQ** variants of our method, as this is the setting that minimizes the IoU of the $q$ output on the 20 DE tiles for both variants.

When training on NY and PA jointly ("Chesapeake" in Table 1), we use the per-state co-occurrence matrices. This ensure that the co-occurrence matrices used are consistent between our method and the self-epitomic LSR benchmark across all columns in Table 1.

**Generalization across cities.** For the high-resolution model with NAIP imagery from Pittsburgh as input, we consider learning rates in $\{10^{-2}, 10^{-3}, 10^{-4}, 10^{-5}\}$ and pick based on the best validation performance on the validation set in Pittsburgh. The chosen learning rate is 1e-3. We search over the same set of learning rates for the model with NAIP imagery and the prior concatenated as input; the chosen learning rate is also 1e-3. For this model with concatenated image and prior as input, only the number of input channels changes in the fully connected network model architecture. When training on the high-resolution land cover labels, we use a very small additive constant 1e-8 for the last layer of the model.

When training our method, we initialize model weights using the best NAIP image input model from the Pittsburgh validation set runs, and then train using the priors and the training procedure described in the main text. We pick the learning rate for this training step using again the validation set in Pittsburgh; we search learning rates in $\{10^{-3}, 10^{-4}, 10^{-5}\}$, and pick 1e-5 as the learning rate, since

resulted in the best performance for both **QR** and **RQ** in the Pittsburgh validation set. We discuss the results of a similar procedure using randomly initialized model weights in Appendix D.1.4.

For the learned prior, we use a 3 layer fully connected network is kernel sizes of 11,7, 5 respectively, 128 filters per layer and leaky ReLUs between layers. For each city, we train this model on the prior inputs (blurred and remapped NLCD, roads, buildings, waterways, and water bodies) using a validation set of 5 tiles separate from from the 10 evaluation tiles in each city. We considered learning rates in $\{10^{-3}, 10^{-4}, 10^{-5}\}$ for learning the prior in each city, and chose 1e-4 as it gave most often resulted in the highest accuracies of each validation set. For learning *on* this learned prior, we searched in a condensed space of learning rate in$\{10^{-4}, 10^{-5}\}$, and chose 1e-5 for evaluation runs in each city based on validation set performance in Pittsburgh, PA.

Table D.1: Supplementary results to accompany Table 2.

| Train region | Model | Pittsburgh, PA | | Durham, NC | | Austin, TX | | Phoenix, AZ | |
| | | acc % | IoU % | acc % | IoU % | acc % | IoU % | acc % | IoU % |
|---|---|---|---|---|---|---|---|---|---|
| Pittsburgh | HR | 89.55 | 70.03 | 72.87 | 34.83 | 71.86 | 36.81 | 6.69 | 7.49 |
| (supervised) | HR + aux | 89.48 | 70.75 | 78.86 | 47.67 | 77.20 | 50.62 | 66.04 | 27.92 |
| Same as test | **QR** ($q$) | 80.46 | 56.81 | 77.77 | 43.92 | 79.59 | 51.93 | 72.20 | 21.86 |
| (random | **QR** ($r$) | 80.62 | 57.44 | 77.91 | 44.86 | 79.59 | 52.30 | 72.39 | 22.32 |
| initialization) | **RQ** ($q$) | 51.74 | 10.35 | 70.63 | 26.83 | 74.32 | 45.99 | 66.96 | 19.16 |
| | **RQ** ($r$) | 2.95 | 0.59 | 70.63 | 26.83 | 74.33 | 46.00 | 66.96 | 19.16 |
| Same as test | **QR** ($q$) | 85.41 | 64.71 | 78.90 | 46.56 | 79.28 | 51.67 | 73.21 | 39.09 |
| (pretrained | **QR** ($r$) | 85.38 | 65.74 | 79.09 | 48.80 | 79.63 | 53.01 | 73.44 | 40.62 |
| in Pittsburgh) | **RQ** ($q$) | 87.35 | 67.88 | 78.80 | 44.43 | 78.65 | 51.90 | 70.59 | 34.94 |
| | **RQ** ($r$) | 87.45 | 68.59 | 78.83 | 44.61 | 78.79 | 52.47 | 70.70 | 36.10 |
| Same as test | **QR** ($q$) | 87.21 | 68.54 | 78.74 | 47.35 | 79.83 | 51.10 | 74.57 | 39.59 |
| (learned prior ) | **QR** ($r$) | 86.97 | 69.27 | 78.93 | 49.57 | 80.27 | 52.60 | 75.31 | 42.22 |
| Full US* [42] | U-Net Lrg. | 79.51 | 61.62 | 77.35 | 48.78 | 76.40 | 51.35 | 24.49 | 23.21 |

### D.1.4 ADDITIONAL RESULTS

Table D.2: Comparison of the Full US* U-Net Large [42] map predictions when evaluated on the full 5 classes considered in Table 2 (water, grass/field, trees/shrub, impervious surfaces, and barren land) and evaluated on the four prediction classes predicted by the model (where barren land and impervious surfaces are merged as a single class), and when barren is post-facto assigned whenever the predicted class is "impervious surfaces" and the label class is "barren land".

| Classication Scheme | Pittsburgh, PA | | Durham, NC | | Austin, TX | | Phoenix, AZ | |
| | acc % | IoU % | acc % | IoU % | acc % | IoU % | acc % | IoU % |
|---|---|---|---|---|---|---|---|---|
| 5 Classes | 79.29 | 55.01 | 76.86 | 42.62 | 76.05 | 48.80 | 18.06 | 18.49 |
| 4 Classes | 79.29 | 68.76 | 77.35 | 53.13 | 76.40 | 59.84 | 24.49 | 16.33 |
| Barren reassigned | 79.51 | 61.62 | 77.35 | 48.78 | 76.40 | 51.35 | 24.49 | 23.21 |

**Extended results for generalizing across EnviroAtlas cities.** The extended results for generalizing across cities with the EnviroAtlas datasets in Table D.1 contain the results of the **RQ** runs trained on the handmade prior in each city, and additionally shows the evaluation results in Pittsburgh, PA, to give further context full comparison of generalization across cities by each method.

Table D.1 also details the result of initializing the model weights randomly for the **QR** method. For this experiment, we again search over learning rate in [1e-3,1e-4,1e-5] in Pittsburgh, this time resulting in a choice of 1e-4 for the **QR** evaluation runs and 1e-3 for the **RQ** evaluation runs. This seems to be a suboptimal choice of learning rate for the **RQ** method in other cities (see Table 2), so the most fruitfull comparison between initialization schemes is between the **QR** variant. Table D.1 shows that the choice of model initialization can be important for our method – this is most apparent

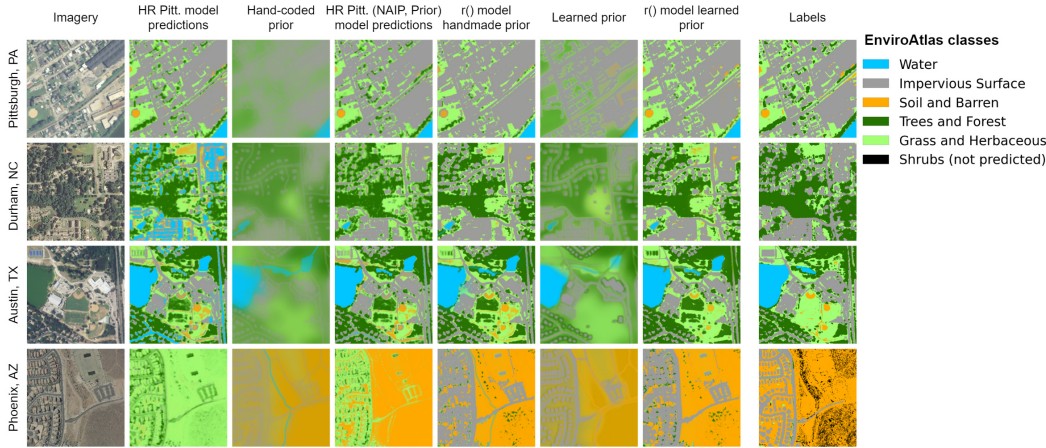

Figure D.2: Example predictions on the hand-coded and learned prior in each EnviroAtlas city we study.

in Pittsburgh, PA (unsurprisingly since the high-resolution model was trained in Pittsburgh) and Phoenix, AZ. In Phoenix, much of the handmade prior is consistent across geographies and the randomly initialized model has trouble distinguishing between infrequent classes that most often occur together in the handmade prior. The results in Table D.1 suggest that using pre-trained models as a starting point for our method can help to break some of these symmetry issues in resolving the information in the prior. Results in Table 2 suggest that using a more detailed prior map may help with this as well.

**Evaluating the Full US map from [42].** Recall that the row for the full US Map [42] in Table 2 reflects the performance of the model evaluated on all 5 classes we consider in our experiments, where we give the map predictions the "benefit of the doubt" in that any prediction of "impervious surfaces" where the true label is "barren land" gets assigned a correct classification of "barren land." The results reported in Table 2 are thus a sort of upper bound on the predictive performance of the method that generated the predictive maps. It was important for us to keep the barren class while evaluating across cities, as it is the dominant class in Phoenix, AZ. In the remaining three cities, the barren class is challenging to predict as it is infrequent. In Table D.2, we compare this classification scheme with two alternatives: a 5 class scheme that will penalizes the map predictions for never predicts the barren class, and a 4 class scheme that merges the barren land and impervious surfaces classes in evaluation. Table D.2 shows that while the choice of evaluation scheme does not greatly effect accuracy (outside of Phoenix, AZ, where the accuracy of the Full US Map is low for both classification schemes), the average IoU drops significantly for all cities apart from Phoenix.

**Comparing loss functions: qualitative results with land cover mapping.** Figure D.3 compares predictions under different loss functions with an illustrative example. Here the prior is similar to the "hand-coded" prior described in Appendix D.1.2, but with the prior defined over all NLCD classes. We train each model (a slight variant on the network used in experimental results) on the single NAIP tile region encompassing the zoom-in in the figure for 2000 iterations with the Adam algorithm [20], a batch size of 64, and a learning rate fixed at 1e-4 during training. Qualitative comparisons show that predictions made by the **QR** and **RQ** loss functions are more certain (sharper colors in plots) than training with cross entropy or squared-error loss on the soft priors, and, in in most places, arrive at better solutions than training with a standard cross entropy loss on the argmax of the prior.

# E ADDITIONAL EXPERIMENTS

## E.1 SELF-SUPERVISION FOR UNSUPERVISED IMAGE CLUSTERING

Neural networks are usually trained on large amounts of hard-labeled data $\{x_i, \ell_i\}$, yet, due to the biases induced by the typical architectures and learning algorithms, a lot of the modeling power of

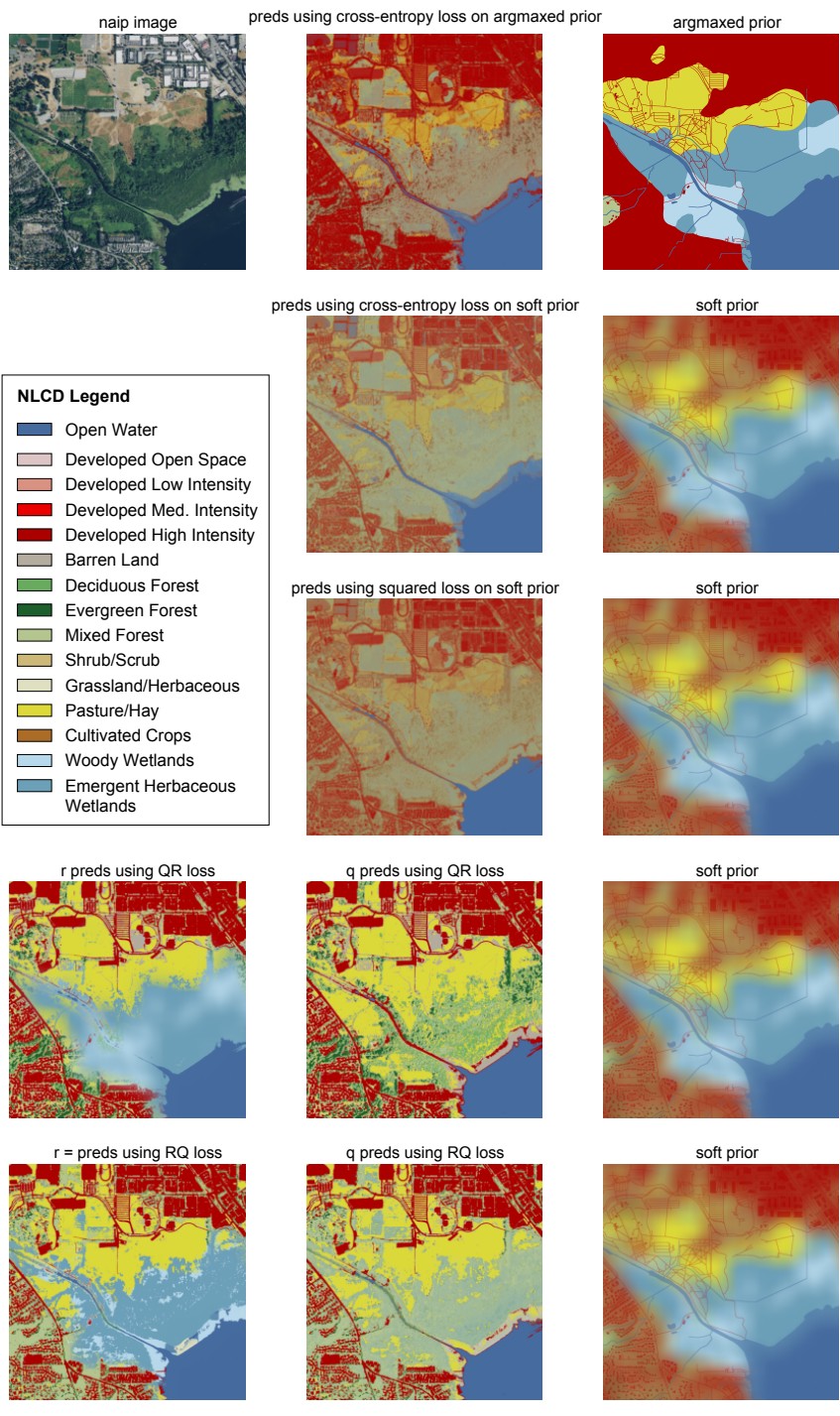

Figure D.3: Comparison of different loss functions on hard and soft prior.

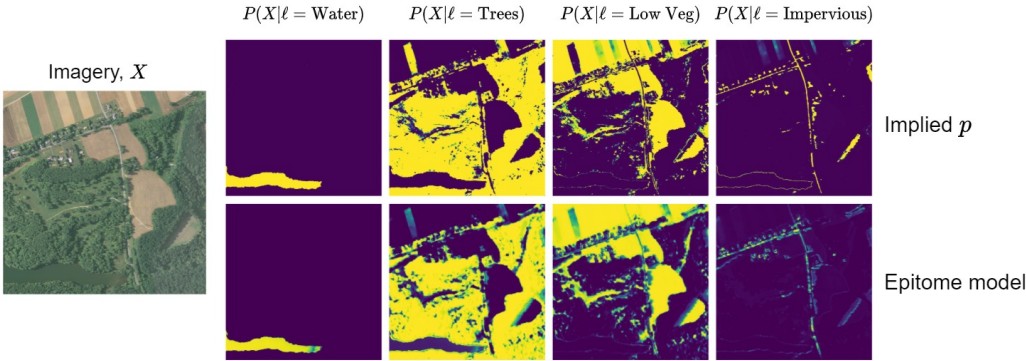

Figure D.4: Comparison of forward model likelihoods under the implicit generative model trained with **QR** loss (above) and the likelihood under an epitome model [28] for part of a test tile from §4.3.

these networks seem to focus on correlations in the input space [46]. This means that a network trained for one application, i.e., for one label space $\ell \in L_1$, can be adopted to another application, i.e., a different labels space $\ell \in L_2$, as long as the input features are in a similar domain. The canonical example of this is the use of lower levels of the networks pre-trained on ImageNet as part of the networks solving a completely different set of image classification problems. Pretrained networks require smaller training sets in fine tuning, as long as they have learned to represent the variation in the input space well. Self-supervised models attempt to go a step further and learn these representations without any labels. In our framework, self-supervision can simply be seen as the appropriate choice of subset priors $p(\ell_T)$ over appropriately chosen tuples of labels.

To discuss the pitfalls and opportunities, consider the **QR** loss (4)

$$F = -\sum_{i,\ell} q_i(\ell) \log p_i(\ell) - + \sum_{i,\ell} q_i(\ell) \log \sum_j q_j(\ell). \tag{6}$$

If we were to simply set $p_i(\ell)$ to a constant (e.g., uniform) distribution $p(\ell)$ for all data points $i$, then the optimal solution would be any function $q_i(\ell) = q(\ell|x_i)$ such that $\frac{1}{N} \sum_i q(\ell|x_i) = p(\ell)$. Thus simply using the uniform prior may not lead to appropriate unsupervised clustering (or self-supervised learning of the network $q$). The inductive biases in the network architecture and training may not help, because one solution is $q(\ell|x) = p(\ell)$, which can be achieved by zeroing out all weights except for biases in the last softmax layer that outputs probabilities for labels $\ell$. As the softmax bias vector is the closest to the top in gradient descent, it will quickly be learned to match $\log p(\ell)$ and this will not only slow down the propagation of gradients into the network, but can eventually stop it completely, as this solution is a global optimum. Another optimal solution would be a function satisfying $\frac{1}{N} \sum_i q(\ell|x_i) = p(\ell)$, but where individual entropies for each data point are small: $-\sum_\ell q(\ell|x_i) \log q(\ell|x_i) < \epsilon$, which motivates an alternative cost criterion

$$F = -\sum_{i,\ell} q_i(\ell) \log q_i(\ell) + \sum_{i,\ell} q_i(\ell) \log(\sum_j q_j(\ell)). \tag{7}$$

where the first term promotes certainty in predictions $q(\ell|x_i)$ for each point $i$ and the second is promoting the diversity of the predictions across the different inputs, i.e., a high entropy of the average $\frac{1}{N} \sum_h q_i(h)$. This prevents learning a network with a constant output $q(h) = p(h)$ and forces the model to find some statistics in the input data that break it into clusters indexed by labels $\ell$. The result will be highly dependent on the inductive biases associated with the network architecture and SGD method used, as we can imagine degenerate solutions here as well. For example, we can ignore completely some subset of features and still train a network that is certain in its modeling of the remaining ones, and achieves a high diversity of predicted classes across the dataset. Obviously, this may be dangerous if the features omitted end up being the most important ones for the downstream task. However, due to the stochastic gradient descent training as well as their architecture, it has been difficult to prevent neural networks from learning statistics involving all the input features. For example, training a neural network using a weak generative model as a teacher corresponds to

using a simpler mixture model, whose posterior is used as a target $p_i(\ell)$ and then learning a neural network that can approximate it. The inductive bias then leads to networks that do not match $p_i(\ell)$ exactly but learn more complex statistics instead.

Equation (7) can be seen as a degenerate example of using a tuple prior where the tuple has the same data point repeated and the prior simply expects the two predictions to be the same. In many applications, there are natural constraints involving multiple data points that are easily modeled with priors over tuples or over the entire collection of labels. Consider unsupervised image segmentation, for an example. It is usually expected that nearby pixels should belong to the same class (or a small subset of classes), and that faraway pixels are more likely to belong to a different subset of classes. This belief is typically modeled in terms of Markov random field models of joint probabilities of labels in the image,

$$p(\{\ell_i\}) \propto \exp \sum_i \phi(\ell_i, \{\ell_j\}_{j \in N_j}). \tag{8}$$

We experimented with potentials of the form

$$\phi(\ell_i = \ell, \{\ell_j\}_{j \in N_j}) = \gamma_\ell + \alpha_\ell \frac{1}{|S_i|} \sum_{j \in S_i} \mathbb{1}[\ell = \ell_j] + \beta_\ell \frac{1}{|L_i|} \sum_{j \in L_i} \mathbb{1}[\ell = \ell_j], \tag{9}$$

where for pixel $i$, $S_i$ is a small ($5 \times 5$) neighborhood around it and $L_i$ is a larger ($50 \times 50$) neighborhood. If we set $\alpha_\ell = 1$, $\beta_\ell = -1$ for all $\ell$, then we consider this a contrastive prior, as it favors labels $\ell_i$ to match the labels found more concentrated in its immediate neighborhood than in the larger scope. On the other hand $\alpha_\ell$, and $\beta_\ell$ can be estimated based on the current statistics in the label distribution using logistic regression, and we refer to this as a self-similarity prior $p(\{\ell_i\}; \alpha_\ell, \beta_\ell, \gamma_\ell)$ with parameters which are periodically fit to the current statistics in the predictions $\sum_{j \in S_i} q(\ell|x_j)$, and $\sum_{j \in L_i} q(\ell|x_j)$ to promote similar label patterns across the image. The criterion (7) can also be seen as a degenerate version of this setting with $S$ being $1 \times 1$ and $L$ being infinite (or the whole image).

The contrastive version of this prior relies on the insight previously pursued in image self-supervision, e.g., [14]. However, contrasting is accomplished without sampling triplets, but considering all the data jointly, simply by expressing the goal of contrasting with far away regions within the prior in our framework.

As an example of self-supervised pretraining in our framework, in Fig. E.1 we show an example of clustering of a large tile of aerial imagery into 12 classes using 5 layer FCN as network $q$ of the architecture used in §4.4. The clustering is achieved by updating the prior every 50 steps of gradient descent on batches of 200 $256 \times 256$ patches. The prior is initialized to a contrasting prior, and then updated through gradient descent. After 7 iterations, the result is sharpened by continuing training using (7).

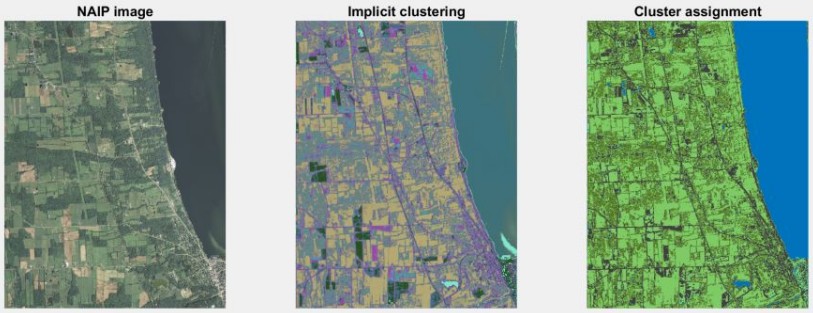

Figure E.1: Unsupervised clustering using implicit **QR** loss (middle) of a NAIP tile (left). On the right, we show the assignment of the 12 clusters to 4 land cover labels: water (blue), tall vegetation (darker green), low vegetation (lighter green) and impervious/barren (gray).

This tile was recently used in testing the fine-tuning of a pretrained model with minimal amount of new labels in a new region [43]. Both the pre-training region, the state of Maryland, and the testing region, the tiles in New York State, come from the 4-class Chesapeake Land Cover dataset (§4.3).

Yet, the slight shift in geography results in reduction of accuracy from around 90% in Maryland down to around 72.5% in New York. In [43], various techniques for quick model adaptation are studied, on labels acquirable in up to 15 minutes of human labeling effort per tile. In Table E.1 we compare the tunability of our self-supervised models on the four 85km$^2$ regions tested in [43] with active learning approaches to tuning a pre-trained Maryland model with 400 labeled points. We show in the table the accuracy and mean intersection over union from [43] for tuning the pretrained model's last $64 \times 4$ layer with different active learning strategies for selecting points to be labeled. A random selection of 400 points for which the labels are provided yields an average accuracy improvement from 72.5% to 80.6%.

On the other hand, recall that we have created an unsupervised segmentation into 12 clusters, with posteriors over the clusters $q_i(\ell)$. To investigate how well these clusters align with ground truth land cover labels, we compute a simple assignment of clusters to land cover labels. Given a set of labeled points $\{(i, c_i)\}_{i \in I}$, we infer a mapping from clusters to four target labels,

$$p(c|\ell) \propto \sum_{i \in I : c_i = c} q_i(\ell).$$

The label of any point $j$ can now be inferred as $\hat{\ell}_j = \arg\max_c \sum_\ell q_i(\ell) p(c|\ell)$. This procedure, using 400 randomly selected labeled points, yields an average accuracy of 81.1% (averaged over 50 random collections of labeled points), which is above the performance of the pretrained model tuned on as many randomly selected points, and on par with the more sophisticated methods for point selection and the use of the pretrained model. (Note that the large model pretrained was trained on a large similar dataset in a nearby state). (Table E.1).

Table E.1: Finetuning a pre-trained model by gradient descent [43] versus implicit **QR** clustering + label assignment in low-label regimes.

| Query method | pretrained model in [43] | | | | Implicit QR |
| --- | --- | --- | --- | --- | --- |
| | No tuning | Random | Entropy | Min-margin | Random |
| Tuned parameters | 0 | 64×4 | 64×4 | 64×4 | 12×4 |
| Accuracy % | 72.5 | 80.6 | 73.6 | 81.1 | 81.1 |
| IoU % | 51.0 | 60.8 | 50.1 | 60.8 | 59.8 |

### E.2 TUMOR-INFILTRATING LYMPHOCYTE SEGMENTATION

The setup of this experiments mimics that of the land cover label super-resolution experiment in §4.3. The training data consists of 50k $240 \times 240$ crops of H&E-stained histological imagery at $0.5\mu$m/px resolution, paired with coarse estimates of the density of tumor-infiltrating lymphocytes (TILs) created by a simple classifier, at the resolution of $100 \times 100$ blocks. The goal is to produce models for high-resolution TIL segmentation. Models are evaluated on a held-out set of 1786 images with high-resolution point labels for the center pixel.

The coarse density estimates $c$ belong to one of 11 classes, from 0 (no TILs) to 9 (highest estimated TIL density). We use an estimated conditional likelihood $p(\ell|c)$ of the likelihood of the positive TIL label at pixels with each low-resolution class $c$ to construct a prior $p_i(\ell)$ over the TIL label probability. Notice that this prior is the same for all pixels in any given low-resolution, coarsely labeled block.[4]

We train a small CNN with receptive field $11 \times 11$ (five ReLU-activated convolutional layers with 64 filters) under the **RQ** loss against this prior for 200 epochs with learning rate $10^{-5}$, then evaluate on the held-out testing set. Inspired by [28], we apply a spatial blur of 11 pixels to the predicted log-likelihoods (again correcting for the model's small receptive field and the dataset bias).

---

[4]We experimented with setting $p_i(\ell|c)$ to conditional likelihoods estimated from a held-out set and with simply setting $p_i(\ell = 1|c = 0) = 0.05$, $p_i(\ell = 1|c = 1) = 0.15$, ..., $p_i(\ell = 1|c = 9) = 0.95$. The latter gave better results, perhaps due to the bias of the evaluation set, in which every image is known to be centered on a cell of some kind.

Table E.2: Area under ROC curve for various predictors on the TIL segmentation task.

| Model | fully supervised | | | weakly supervised | | |
| | SVM [58; 13] | CNN [13] | CSP-CNN [13] | U-Net [27] | Epitome [28] | **RQ** |
|---|---|---|---|---|---|---|
| AUC | 0.713 | 0.494 | 0.786 | 0.783 | 0.801 | 0.802 |

The AUC scores of this model and of the baselines are shown in Table E.2. Interestingly, the best-performing models – **RQ** and epitomic super-resolution (a generative model) – both have receptive fields of $11 \times 11$, much smaller than those of the U-Net and fully supervised CNNs. This means that prediction of TIL likelihood is possible using only *local* image data, but the challenge is learning to resolve highly uncertain label information. Unlike U-Nets and deep CNN autoencoders, small models are not able to learn and overfit to *distant* spurious clues to the classes of nearby pixels.

### E.3 VIDEO SEGMENTATION WITH A STRUCTURED PRIOR

To demonstrate the use of priors with latent structure, we set up the problem of video segmentation as follows. Given a frame $t$, we tune networks $q_t(\ell_{i,t}|x_{i,t})$ predicting one of $L$ pixel classes for a pixel at coordinate $i$ in frame $t$. The prior in each frame comes from a Mask R-CNN model [11] pre-trained on still images in the COCO dataset [23]. The Mask R-CNN model finds several possible instances of objects of different categories and outputs the soft object masks in form of confidence scores for each pixel. We convert this into a probability distribution over the index $f$ (foreground/background) of the form $p(f_{i,t}|m_t)$, where $m_t$ are different detected instances by the model, and the distributions $p(f_{i,t}|m_t)$ are the soft masks for these instances converted to probability distributions, i.e. value of the probability of foreground differs for each pixel and each instance based on the Mask R-CNN confidence scores. Although the COCO dataset may not have had instances of object of interest in our frame $x_t$, we assume that some admixture of detected instances (likely involving unrelated types of objects) does model reasonably well the foreground segmentation in the frame. Mathematically, $p(f_{i,t}) = \sum_{m_t} p(f_{i,t}|m_t)p(m_t)$, where $p(m_t)$ expresses the probabilistic selection of the foreground masks for different instances from which the foreground is constructed. (One can think of instances $m_t$ as topics in topic models, which are also admixture models). To complete the prior, we fix the distribution $p(\ell|f)$ as fixed binary $L \times 2$ matrix assigning a subset of $L$ pixel classes to foreground and the rest to the background. (For example, we assign first 3 classes to foreground and the remaining 5 to the background for a total of L=8 pixel classes). Therefore,

$$p(\ell_{i,t} = \ell) = \sum_f p(\ell|f)p \sum_{m_t} p(f_{i,t} = f|m_t)p(m_t) \tag{10}$$

We can now select the instances $m_t$ in each frame by optimizing the free energy with this prior over $p(m_t)$. The procedure involves standard variational inference of the posterior distribution over possible instances $m_t$ for each pixel $i$ in frame $t$ which involves the posterior $q_t(\ell_{i,t}|x_{i,t})$. In practice we found that it is enough to do this inference once, using the network $q_{t-1}$ estimated in the previous frame.

This requires the inference of $m_t$ for each pixel $i$:

$$s_i(m_t) \propto \exp(\sum_i \sum_{\ell,f} p(\ell|f)q_t(\ell_{i,t} = \ell|x_{i,t}) \log p(f_{i,t} = f|m_t)p(m_t), \tag{11}$$

and then optimizing $p_{m_t}$ as the count of times each instance is used,

$$p(m_t) \propto \sum_i s_i(m_t) \tag{12}$$

Selection of instances $m_t$ in frame $t$ therefore involves comparing the predictions from the network $q_t(\ell_{i,t} = \ell|x_{i,t})$ grouped into foreground/background segmentation with the foreground/background segmentation for different instances from Mask R-CNN, and making a selection of a subset (probabilistically in $p(m_t)$) based on which instances most overlap with the predictions from network $q_t$. While the above two equations should in principle be iterated, and iterated with updates to network $q_t(\ell_{i,t} = \ell|x_{i,t})$, we found that in practice it is sufficient to just select the instances $m_t$ based on their

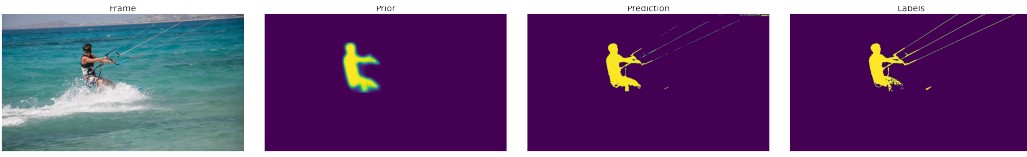

Figure E.2: Example of inferring the foreground mask on a video frame.

intersection with the network predictions once, at the very beginning, to make a soft fixed prior, and leave it to optimizing the prediction network with the **RQ** loss to find confident segmentation (Fig. E.2).

We tested the approach on the DAVIS 2016 dataset [37]. The dataset is comprised of 50 unique scenes, accompanied by per-pixel foreground/background segmentation masks. The objective is to produce foreground segmentation masks for all frames in a scene, given only the ground truth annotations of the first frame (Semi-Supervised). We evaluated our method on the 20-scene validation set at 480p resolution.

The network $q$ used in this experiment combines both the pixel intensities and spatial position information for its predictions. At each pixel location $i, j$, we augment the intensity information with learned Fourier features $[\sin(W[i, j]^T), \cos(W[i, j]^T)]^T$ [47]. The image and spatial position are first processed separately; A 4-layer, 64-channel, fully-convolutional network with $3 \times 3$ kernels, ReLU activations and Batch Normalization produces the image features. A 3-layer, 16-channel, pixel-wise MLP with ReLU activations and Batch Normalization processes the learned Fourier features. These two are concatenated and passed through a single $3 \times 3$ convolution-ReLU-Batch Normalization layer before being mapped to output predictions. We also experimented with adding optical flow as another auxiliary input to the network.

For each scene, the network $q_0$ is trained on the first frame, using the given ground truth annotations split uniformly between 3 foreground and 5 background classes as prior, for 300 iterations. This network is then used to predict the foreground pixels in the next frame and after computing the intersection over union between the predicted foreground pixels and the Mask R-CNN output masks, we select masks that overlap more than a pre-specified threshold. The chosen masks are then summated, weighted by their Mask R-CNN confidence scores (0-1), to form the prior for the next frame. The process of selecting masks from the Mask-RCNN predictions and forming the prior for a frame is showcased in Figure E.3. The network $q_0$ is then fine-tuned for 10 iterations to obtain $q_1$ and this process repeats for all subsequent frames. We used the Adam optimizer, with a starting learning rate of $10^{-3}$ for the first frame, reduced to $10^{-5}$ for fine-tuning, and trained with batches of 128 64×64 patches.

To infer the foreground pixels we first need a pre-trained the Mask R-CNN on the COCO dataset. Then, for each scene we only require ~1m of training time on the ground truth-annotated first frame and ~3s per every following frame for the entire process of forming the prior and inferring the foreground pixels. We do not train on any video data, in contrast to most video object segmentation methodologies that rely on both a pre-trained network on static image datasets (such as COCO) and additionally on offline training on video sequences. In Table E.3 we compare our results on the DAVIS 2016 validation set to other video object segmentation algorithms from 2017 - present.

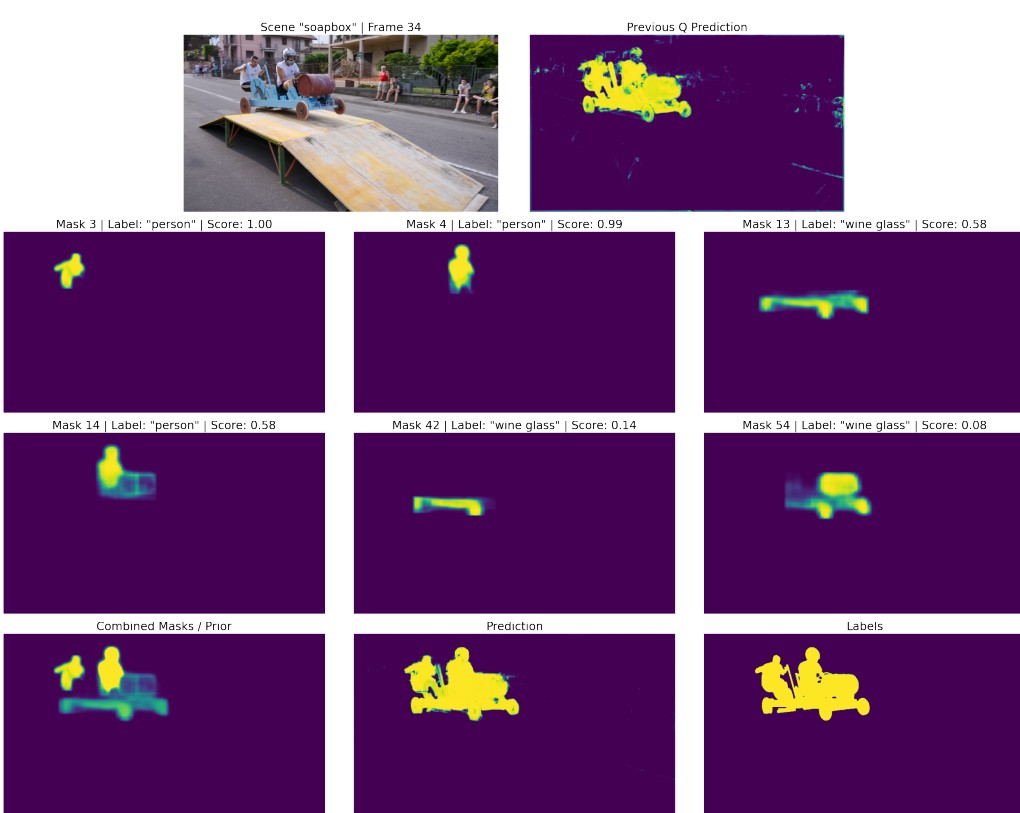

Figure E.3: Video segmentation procedure. Starting with a network $q_{t-1}$ trained on frame $t-1$, we apply $q_{t-1}$ on frame $t$ to get a rough foreground estimation (top). By running the pre-trained Mask R-CNN model on frame $t$ and selecting only the masks that overlap with the $q_{t-1}$ prediction we get the candidate object masks (middle). The prior is constructed as the sum of the candidate masks, weighted by their corresponding Mask R-CNN scores, and $q_{t-1}$ is finetuned on frame $t$ with this prior to get the detailed predictions (bottom).

Table E.3: Jaccard and F1 measures for various algorithms on the video instance segmentation task.

| Model | J&F ↑ | J Mean ↑ | J Recall ↑ | J Decay ↓ | F Mean ↑ | F Recall ↑ | F Decay ↓ | Year |
|---|---|---|---|---|---|---|---|---|
| OSVOS [3] | 80.2 | 79.8 | 93.6 | 14.9 | 80.6 | 92.6 | 15 | 2017 |
| MSK [38] | 77.55 | 79.7 | 93.1 | 8.9 | 75.4 | 87.1 | 9 | 2017 |
| OnAVOS [49] | 85.5 | 86.1 | 96.1 | 5.2 | 84.9 | 89.7 | 5.8 | 2017 |
| Lucid [18] | 82.95 | 83.9 | 95 | 9.1 | 82 | 88.1 | 9.7 | 2017 |
| OSVOS-S [29] | 86.55 | 85.6 | 96.8 | 5.5 | 87.5 | 95.9 | 8.2 | 2018 |
| FAVOS [5] | 80.95 | 82.4 | 96.5 | 4.5 | 79.5 | 89.4 | 5.5 | 2018 |
| PReMVOS [25] | 86.75 | 84.9 | 96.1 | 8.8 | 88.6 | 94.7 | 9.8 | 2018 |
| OSMN [52] | 73.45 | 74 | 87.6 | 9 | 72.9 | 84 | 10.6 | 2018 |
| AGAME [16] | 81.85 | 81.5 | 93.6 | 9.4 | 82.2 | 90.3 | 9.8 | 2019 |
| STM [36] | 89.4 | 88.7 | 97.4 | 5 | 90.1 | 95.2 | 4.2 | 2019 |
| FEELVOS [50] | 81.65 | 81.1 | 90.5 | 13.7 | 82.2 | 86.6 | 14.1 | 2019 |
| CFBI [53] | 89.4 | 88.3 | - | - | 90.5 | - | - | 2020 |
| e-OSVOS [30] | 86.8 | 86.6 | - | - | 87 | - | - | 2020 |
| STCN [4] | 91.7 | 90.4 | 98.1 | 4.1 | 93 | 97.1 | 4.3 | 2021 |
| Ours | 83.8 | 84 | 96.2 | 8.4 | 83.6 | 94.2 | 10.2 | |
| Ours (+flow) | 83.9 | 83.2 | 95.5 | 9.5 | 84.6 | 93.3 | 9.1 | |

### E.4 SEDUCERS SEDUCING SEDUCERS

One of the conclusions from our experiments on EnviroAtlas (§4.4) is that training a network with the goal of generalizing to new input data is often inferior to simply performing in-collection inference. In other words, given the collection of pairs $x_i, p_i(\ell)$, learning the posterior $q$ under the implicit generative model is optimized for resolving ambiguities in that collection, and possibly that collection alone. As pointed out in [28], which performs collection inference using large generative models to mine self-similarity among the examples in the collection, this is appropriate when we can expect our data $x_i$ to always come paired with prior beliefs $p(\ell_i)$. It is interesting to reconsider the Seducer example from Fig. 1. The artist created several versions of that painting, and as shown in Fig. E.4 collection inference applied separately to each of these painting works equally well. However, using a learned $q$ network from one image onto others yields inferior segmentations (Fig. E.5), as the learned network specialized for inference in the data it saw. (A fully generative model would similarly overtrain on the input data features $x_i$, as would a supervised neural network trained on hard-labeled pairs $(x_i, \ell_i)$ due to the domain shift.) Yet, if we know we will always be given collections with beliefs in the form of priors $p_i(\ell)$, local (collection) inference is all we need.

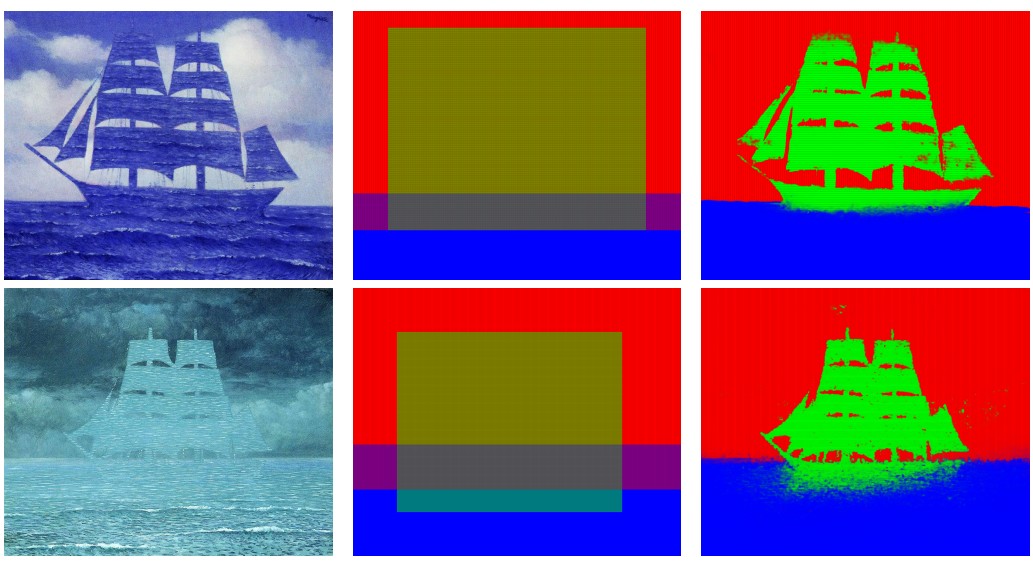

Figure E.4: Two additional versions of *Le séducteur* (left), hand-made priors (middle) and inferred segmentations (right).

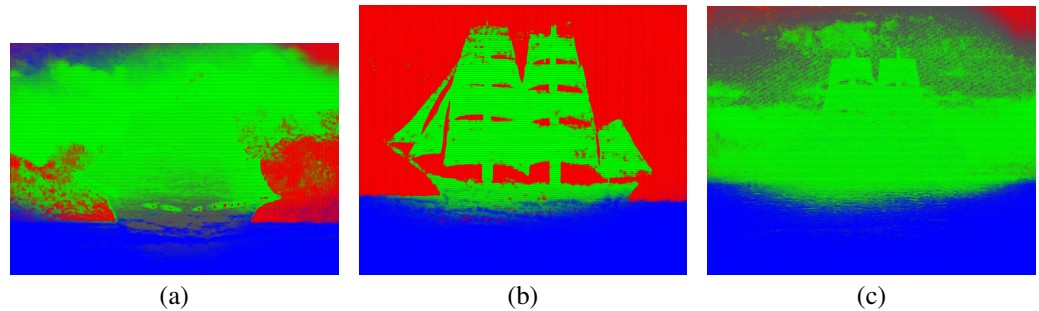

| (a) | (b) | (c) |

Figure E.5: Result of applying a network $q$ trained to infer (b), on all three *Le séducteur* versions.

