# OpenReview forum: "Resolving label uncertainty with implicit generative models"
_ICLR.cc/2022/Conference — ICLR 2022 Submitted_

### Official Review · Reviewer_HHeb · 2021-11-02

**Correctness:** 3
**Technical Novelty And Significance:** 3
**Empirical Novelty And Significance:** 3
**Recommendation:** 6
**Confidence:** 4

**Main Review:**

Strength:

+The idea of implicit generative models with label priors to handle coarse and imprecise data input is interesting and easy to follow.

+The experiment validation covers five different tasks in both CV and NLP fields, which shows a potential good and positive impact for implicit generative models to the CV and NLP communities.

Weakness:

-The paper is not well written and organized. Especially in the introduction section, the background of the problem and the motivation of introducing implicit generative models are not well stated. I  suggest the authors to move Section A in the Appendix into the Section 2 as a separative subsection.

-The experiment validation still has room to improve. As the goal of this paper is to resolve label uncertainty, the authors should compare the proposed methods with some state-of-the-art approaches like [35; 54; 53; 55; 58] to make the experiment validation more solid.

-It should be better if the authors can provide discussion on limit and failure cases of the proposed method existing in different tasks.


**Summary Of The Paper:**

In this paper, the authors present implicit generative models in a free energy criterion with a combination of both the training of neural networks for label prediction and modeling of a label prior. They also discuss multiple sources of label priors to handle label uncertainty for coarse and imprecise data input. The extensive experiments conducted on five different tasks and the experimental results seem to well validate the efficacy of the proposed method.

**Summary Of The Review:**

The idea is interesting and easy to follow. It has a very positive impact to the AI community. However, the paper writing and organization, experiment design and presentation still have room to improve. Therefore, I initially tend to marginally accept this paper.

---

> ### Author Response · Authors · 2021-11-18
> **Response**
>
> Thank you for your suggestions to improve the clarity of our writing in the introduction and background section. We will revise these sections, and following your suggestion will move relevant discussion in section A to the main text to aid in exposition and clarity for our readers.
>
> Regarding the suggested comparison with citations [35; 54; 53; 55; 58], we feel this is another place we can clarify further the introduction. While these references are related to our approach in that they encourage low entropy of predictions (as we state in the paper), direct comparison to each of these approaches either would not make sense in our setting or would, as noted above, only handle a subset of our settings:
> - [58] is about semantic segmentation with image-level labels and relies on a domain-specific augmentation procedure. It cannot readily be extended to, for example, pixel-level priors.
> - [55] is about unsupervised clustering applied to object re-identification, and the approaches do not transfer in a clear way to any of our settings.
> - [35,53,54] are all about partial-label learning and loss functions that decrease prediction entropy, with a focus on image classification. The lack of a generative formulation in these loss functions makes them ill-suited to problems with *varying* uncertainty encoded in priors. (For example, in a two-class setting where some data has a 60% prior on class A and some has a 60% prior on class B, these losses -- unlike ours -- have no incentive not to assign all data to class A and none to class B.)
>
> Our edits to the introduction section will make these points clear, which we believe will help contextualize our contributions for our readers.
>
> Thank you for your suggestion to provide limit and failure cases of the proposed method. We will address these in a "practical considerations" section which we will add to the end of the paper, highlighting also the unifying lessons (and failure modes) learned across the variety of experiment domains we study. The addition of this section will strengthen and delineate the key takeaways of our experimental sections. (Note also that the paper and the appendix do discuss one extreme case -- that of uniform labels everywhere -- and point out that additional assumptions on the data have to be incorporated into a prior that may still seem fairly agnostic, but variable across data points. For example, we use a contrastive prior formulation to build self-supervised networks in the Appendix.)

---

### Official Review · Reviewer_cjDc · 2021-11-03

**Correctness:** 3
**Technical Novelty And Significance:** 3
**Empirical Novelty And Significance:** 3
**Recommendation:** 6
**Confidence:** 4

**Main Review:**

The paper addresses an impressive range of learning settings under the same formulation. The paper is clear and well-presented.

On the positive side, I like how the authors connect fields that are in general not address under the same umbrella frameworks, which makes their approach very complete and appealing.

Also, I wanted to highlight the very thorough experimental design and analysis. The choice of case studies is very adequate and in some cases, inspiring.

On the negative side, I would have liked to have seen a more elaborate discussion of the relationship with existing methods for each of the problems the paper addresses. The literature review lacks a bit and it makes it difficult to assess the overlap with previous works.

Also, although the experimental work is very good, I was missing any form of theoretical analysis of the proposed approach.

**Summary Of The Paper:**

The paper presents an approach base on implicit generative models to address a number of existing (and potentially new) scenarios, including weak labels (noisy or incomplete), likelihoods given by a different prediction mechanism on auxiliary input,
or priors reflecting knowledge about the structure of the problem. The experiment presents a selection of very diverse case studies.

**Summary Of The Review:**

I like the paper and its aim, particularly on the experimental front. There is room for improvement on the theoretical analysis and related work.

---

> ### Author Response · Authors · 2021-11-18
> **Response**
>
> Regarding the literature review, please see the comment to all reviewers. We tried to fit a discussion of relevant literature in each application domain in the relevant sections of the paper. In light of your suggestion we will incorporate a distinct Literature Review section in the appendix to delineate related work generally and across each domain in one place. We believe this will help readers contextualize our contributions as you suggest and to compare our approach to general and domain-specific alternative approaches.
>
> As you mention, our work, while motivated from a theoretical perspective (see also Appendix A), is mainly validated through extensive experimentation. We focused on experimental analysis as we thought this the best way to demonstrate the standout contributions of our approach -- ease of integration with existing (deep) learning models and across a variety of domains. Given the length constraints of the conference submission, there was not space to also do extensive theoretical analysis in this work. We agree that such analysis would be an interesting direction for a future paper, especially in light of the empirical results we evidence in this paper. In light of your suggestion we will move some of this discussion to the main text to fully highlight the theoretical motivations behind our algorithm design. We will also add more discussion in Appendix A  on relationship between implicit generative modeling and variational autoencoders and wake-sleep algorithm. (In both cases, an entire half of the model is erased in favor of simple normalization steps in (2) and (3) with a remarkable lack of consequences, which  raises the question of next steps towards hierarchical implicit generative models, where a similar treatment of uncertain latents happens throughout the network and not only at the final label layer.)

---

### Official Review · Reviewer_Da9G · 2021-11-03

**Correctness:** 3
**Technical Novelty And Significance:** 3
**Empirical Novelty And Significance:** Not applicable
**Recommendation:** 5
**Confidence:** 4

**Main Review:**

Pros:
- This work considers a general form of weakly-supervised problems, which can be applied to a wide range of scenarios, including learning from partial or coarse labels.
- The paper proposes a new strategy by constructing an implicit generative model that incorporates the weak label priors in a unified manner.
- On a range of vision and text classification tasks, the proposed method seems to achieve competitive performances with higher efficiency.

Concerns:
- The proposed implicit generative model seems expensive to compute due to its normalization factor. As shown in Equation 2, the forward probability requires a summation over the entire training dataset, which is also used in the loss in Equation 4. It is unclear how to implement this in a mini-batch mode. Note the summation within the log should be over i instead of l.
- As discussed in the paper, there are potentially different degenerate solutions for the loss function, and it is unclear in which conditions those solutions can be avoided. The strategy of breaking symmetries used in this work seems a bit ad hoc and no general guideline is provided.
-  The motivation for adopting the RQ loss is unclear. Unlike the QR objective, the RQ loss does not correspond to any properly-defined loss function. More importantly, it seems difficult to choose which one to use in a specific task. In this paper, RQ is adopted for partial labels and text classification, while QR is more effective for land cover segmentation.
- The experimental evaluation seems less convincing due to the following reasons:
1) For the partial label task in 4.1, the improvement over a simple NLL baseline is marginal, as shown in Figure 3 top.
2) For the learning from rank task in 4.2, it lacks comparisons with other learning to rank learning objectives.
3) For the land cover segmentation task in 4.3, it also lacks comparisons with a simple baseline trained with pseudo/inferred labeling.
4) For the learned prior task in 4.4, the pre-training seems to play a key role and it is unclear how important the proposed strategy is. The performance is much worse if the network is trained from scratch.
5) For the text classification in 4.5, it lacks detailed ablative study for the proposed model. It is unclear whether the improvement is due to a better prior adopted in this work or the RQ loss.

**Summary Of The Paper:**

The paper presents a weakly supervised learning strategy, which exploits instance labels in a form of label prior distributions for training classifiers. The main idea of this work is to build an implicit generative model from a probabilistic label prediction network, which is then trained with an ELBO loss. The paper applies this framework to several types of weakly supervised learning scenarios, including classification with negative labels or labels from ranking, semantic segmentation and text classification from coarse labels, and other tasks with structured label priors, etc.  The experimental evaluation validates the efficacy of the proposed learning strategy on the aforementioned tasks with comparisons to the prior works.

**Summary Of The Review:**

The proposed implicit generative model seems interesting and achieves good performance for a range of weakly-supervised tasks. However, I am slightly leaning towards the negative side due to the lack of clarity in the model design and missing comparisons in the experimental evaluation.

---

> ### Author Response · Authors · 2021-11-18
> **Response**
>
> Thank you for your questions on the model design. As mentioned in our response to all reviewers, we will be adding a "practical design considerations" section toward the end of the paper, which will address the three points of clarifications you mentioned to strengthen our work. Specifically:
> - We will detail how we implement and estimation of the summation in equation 2 and 4 with mini-batches (interested readers can also find our implementation in the code that will be provided publicly).
> - We will discuss the different methods for breaking symmetries and avoiding degenerate solutions, including pre-training on limited high-resolution labels as in our EnviroAltas experiments, or pre-training on cross-entropy loss on the prior to break symmetries in the model predictions before training on the QR/RQ losses, or self-supervised pretraining as in the Appendix. It is likely that different approaches to breaking symmetry will be appropriate or feasible in different domains, but we agree that delineating a few general (and simple) options will be beneficial.
> - Similarly, we will add some unifying takeaways regarding the relative applicability of the RQ/QR losses in different settings. (See also the response to all above.)
>
> Regarding the comments on experiments, as we noted in the response to all, our goal was to widely demonstrate a unifying approach to dealing with uncertain labels that has advantages of generative models, but avoids parametrizing generation explicitly. Thus, our experiments consisted of both illustrative examples to show broad applicability and specific SOTA results. We are hoping that the main takeaway is not any single application (even when we do beat the SOTA), but the potential for this framework to be used in many domains. With that in mind, we address specific comments on Section 4:
>
> - S.4.1: (1) the improvement over the NLL baseline is often significant: 93% -> 95% is a substantial improvement, even if it is not clear in the context of the full graphs; (2) as we note in the text the RQ loss is equivalent to NLL loss when Q is uniform, which is usually the case late in training.
> - S.4.2: As you note, we are not *learning to rank*, but *learning from ranks* to separate images into ten discrete classes (So in a sense, the approach can be used in learning to rank applications where ranks are coarse, i.e., defined through a set of ranked equivalence classes). We included this experiment mainly to show the wide range of possible learning settings our proposed method can handle, and are not aware of any benchmarks for this label setting.
> - S.4.3: In response to the reviews, we added a ``hard naïve'' baseline experiment in which we use the most likely high-resolution class given the low-resolution label for each pixel in a standard semantic segmentation training setup with inferred labels. We use the same FCN architecture as the QR/RQ experiments and find that this baseline is unable to identify built areas (due to not being able to properly handle the uncertainty inherent in the low resolution labels), resulting in poor mIoU results. Specifically, it achieves a 59.8 mIoU in NY, 63.0 mIoU in PA, and 59.7 mIoU in Chesapeake. These new baseline results are the worst, or tied for the worst out of all methods. We have added these new results as a row in Table 1.
> - S.4.4: We analyze the effect of using the pretrained weights versus a random initialization in Table B.1.
> The dominant comparison for your point is between the Pittsburgh model weights (first row of the table, "Pittsburgh HR") and the QR methods with either random initialization or pretraining. Table 2 and Table B.1 show that in all cities other than Pittsburgh, the QR model trained from scratch (random initialization) outperforms the Pittsburgh high-resolution (HR) model.
> - S.4.5: We do not understand the comment "a better prior adopted in this work or the RQ loss". Past work [32] does not set a prior on labels, but generates pseudo-training data with associated pseudolabels. The goal of this experiment was to illustrate that our approach applied to a weak prior derived from a pretrained language model can be competitive with far more expensive approaches that query the same language model.
>
> Please let us know if you would like use to list more details on the above experiments, or if additional baselines are needed to make the overall point on wide applicability of the approach: When modeling uncertainty in labels, it is much easier to express beliefs/knowledge in a generative direction, and this can be done in many different ways in different applications (thus the many examples on forming priors in the paper), yet a deep neural net is typically a more powerful inverse model, and so marrying the two without explicit generative modeling can have wide practical applications.

---

### Official Review · Reviewer_5Mem · 2021-11-03

**Correctness:** 3
**Technical Novelty And Significance:** 2
**Empirical Novelty And Significance:** 2
**Recommendation:** 3
**Confidence:** 4

**Main Review:**

This paper discussed an approach to take priors into account in machine learning with a slightly different formula, which I have doubts on its correctness. In the experiments, they showed that performance in some tasks can be significantly improved by cleverly choosing class priors. The experiment results look good, but I do not think it fully justifies their algorithm because of the lack of baselines and inappropriate interpretations of an equation.

The authors mentioned a lot of different types of priors, however many of them (including most of those used in their experiments) could be characterized as cost-sensitive learning, a.k.a. taking priors into account in classification problems. In this regard, the citations to cost-sensitive learning literature and comparisons are vastly insufficient. e.g. papers that should be cited include:

Zadrozny et al. Cost-sensitive learning by cost-proportionate example weighting. ICDM 2003

Branco et al. A survey of predictive modeling on imbalanced domains. ACM Computing Surveys 2016 is a recent survey paper on cost-sensitive learning.

They did not compare against any prior work on cost-sensitive classification, which is a major problem that lead me to the decision of rejection. A recent popular general approach to deal with class priors is focal loss, which should be cited and compared as well.

Lin et al. Focal Loss for Dense Object Detection. ICCV 2017

Despite the authors claims, eq. (4) still seems fundamentally flawed in that in the case of 50%/50% 2-class classification problems, the classifier could have entirely flipped one class with another and still optimize eq. (4). I believe what is working is NOT eq. (4), but the practice of applying eq. (4) on mini-batches. Because each mini-batch is much smaller, it becomes difficult to keep the balance in a mini-batch and the multiple resampling of each mini-batch from different epochs may lead to the effect of normalization, similar to the effects that we can see in minibatch regularization in the training of GANs. Hence, I have strong doubts that similar balance-breaking would not necessarily happen so reliably if mini-batches are not used.

The choice between QR and RQ is quite arbitrary in different experiments. Is there any principle about it? Or just trial-and-error?

There is an equation typo in eq. (4), the log-sum term should be log(\sum_i q_i(l)), not log(\sum_l q_i(l)). The interpretation in the subsequent paragraph is although correct in that it favors predicting a high-confidence label.

Minor: The authors should mention that q_i(l) is an energy function (logits) instead of a probabilistic distribution, otherwise they should have \sum_j q_j(l) = 1 and the derivation would be in a different place (e.g. the \sum_j q_j term won't exist).


**Summary Of The Paper:**

This paper discussed an approach to take priors into account in machine learning with a slightly different formula, which I have doubts on its correctness. In the experiments, they showed that performance in some tasks can be significantly improved by cleverly choosing class priors.


**Summary Of The Review:**

This paper discussed an approach to take priors into account in machine learning with a slightly different formula, which I have doubts on its correctness. In the experiments, they showed that performance in some tasks can be significantly improved by cleverly choosing class priors. The experiment results look good, but I do not think it fully justifies their algorithm because of the lack of baselines and inappropriate interpretations of an equation.

---

> ### Author Response · Authors · 2021-11-18
> **Response**
>
> The problem we study in our paper is learning from data with uncertain labels, where this uncertainty is expressed through a prior distribution over the classes for each sample. We believe that this formulation is inherently different from what you consider as "choosing class priors". As such, the problem we study is not directly related to cost-sensitive learning.
>
> Cost-sensitive learning is utilized to train models on datasets with significant class imbalance. In that setting, class priors are defined as a global prior knowledge of the distribution of all labels in the dataset ($p(l)$ instead of $p_i(l)$). In the example of one-stage object detection, given in the focal loss paper, we expect that most locations in a fixed dense grid will belong to the background class and thus the class prior describes this foreground/background imbalance.
>
> In contrast, the priors in our experiments express a belief on which classes each sample belongs to, e.g. all-but-one digits in the $k=1$ negative labels MNIST experiment or multiple land cover types with varying weights in the land cover mapping example.
>
> To further support that cost-sensitive learning is not directly applicable to our problem, we attempted to perform the experiment described in Section 4.1 (training on MNIST digits with $k=1$ negative labels per sample) using a naïve extension of the focal loss for learning with a distribution over labels in place of our QR/RQ losses. The resulting test accuracy of 0.17 is equivalent to training with the standard cross-entropy loss. Cost-sensitive losses will be unable to deal with this problem setting, as reweighting the classifier's mistakes by designing loss functions that consider the overall label statistics is a *dataset-wide* approach, which is not suitable to resolving the *per-sample* uncertainty that we discuss in our paper.
>
> Regarding your comments on mini-batching and the choice between QR/RQ, see our response to all reviewers. We note that mini-batches are not inherent to the success of our algorithm: during experimentation, we ran small-scale experiments without using mini-batches for single images for land cover mapping and the Seducer (boat image) example. In fact, we had concerns that our technique would suffer if normalizations in (2) and (3) were simply performed on minibatches, which are memory and compute efficient, instead of aggregating over the entire dataset. However, it turned out that in all applications minibatches had enough diversity of data to make (2) and (3) fairly accurate even when summation in the denominator is done only over a subset of data present in the current minibatch. Thus, we see it as an advantage that our approach does work with mini-batches. In particular, we find that our method works even with small batch sizes, allowing it to be used with a wide range of training setups. It is quite possible that in some applications, the normalization in equations (2) and (3) would benefit from aggregation across many minibatches, which is also straightforward to implement with forgetting factors. We will discuss this in the 'practical consideration section'.
>
> Thank you for pointing out the typo in equation 4; we noticed that shortly after submitting and have updated the pdf accordingly.

---

> > ### Comment · Reviewer_5Mem · 2021-11-28
> > **Not convinced**
> >
> > Thanks for the rebuttal. However, I am not convinced by those points.
> >
> > 1) By the definition of a prior, x_i cannot be involved. If x_i is involved, it's no longer called a prior on the class.
> >
> > 2) You would need to do a comprehensive study comparing against cost-sensitive learning. The current job is not enough.
> >
> > 3) Regarding whether mini-batch is inherent to the success, the would have required a more comprehensive study.
> >
> > I keep my ratings the same.

---

> > > ### Author Response · Authors · 2021-11-29
> > > **On terminology and relevance**
> > >
> > > 1. We respectfully disagree with the claim that context-dependent distributions cannot be called "priors". In fact, a number of standard examples of Bayesian statistics involve distributions termed "priors" that depend on auxiliary information or earlier inferences. For instance, in analysis of time series, inferred posteriors from a previous time step can become priors for a value or class at the current time step, not unlike our video segmentation example (Appendix E.3); this is the very foundation of Kalman filtering. Furthermore, imprecise measurements or even expert judgments on individual data points have been considered as a source of priors in science for over a century, even predating more standard notions of probability theory. ([H.Poincaré "Calcul des probabilités" (1896)] states Bayes' theorem and derives least-squares regression as a case of model inference under data-dependent Gaussian priors.)
> > > 2. To put terminological differences aside, let us imagine that we had written "supervision in the form of sample-dependent distributions" instead of "priors" throughout the paper. The methods you mention specifically treat *global class priors/imbalance*, which is not what we study in this work. They are not a relevant basis for comparison.

---

### Author Response · Authors · 2021-11-18
**Response to all reviewers**

Many thanks for your comments and suggestions which will strengthen our work in forthcoming revisions. In this comment, we respond to points made by multiple reviewers.

- Reviewers found the breadth of application domains to be a major strength of the paper. It was important to us to show the variety of problem domains captured under our unifying framework, and to evidence that with experiments in many domains. The goal of our experiments was to show the broad applicability of our approach for learning from uncertain labels, rather than to achieve state of the art in a single task (though in some tasks we do!). We chose the benchmarks in each task domain to investigate and contrast algorithmic properties of our solution with existing methods in some cases, and to benchmark with respect to state-of-the-art performance in other cases where we felt that comparison was appropriate. We have shown the broad potential for implicit generative modeling in these different domains, and hope to see this work spark domain-specific instantiations of implicit generative modeling through our loss functions. Such an endeavor for any one of our five main application domains alone (partial label learning, learning from ranks, land cover mapping in different label settings, medical image segmentation, video segmentation) would each require careful benchmarking and thorough domain-specific experimentation constituting a unique research endeavor and publication in its own right. *Notably, each of the works suggested by reviewers as baseline comparisons is applicable only to one or two of these domains.*

- On that note, some reviewers had questions about the practical design decisions we made in implementing our approach across different experiment domains. These questions exposed to us an opportunity to explicitly discuss the practical design considerations of our proposed approach, serving also as a synthesis across the different experimental domains. We will include a ``practical design considerations'' section toward the end of the paper in our revised version, and will address:
  - mini-batch implementations of our approach suitable for deep learning (reflecting what we use in experiments): Our initial implementation did not use mini-batches, but instead occasionally computed the normalization across the entire tile in land cover experiments, but we found that as long as minibatches are large enough to include enough diversity of x_i p_i(l) pairs,the method works when (2) and (3) are applied directly to minibatches, and in addition, it is possible to update necessary statistics as the learning algorithm moves over different minibatches with a suitable forgetting factors.
  - relative benefits/limitations of the QR and RQ loss formulations: The QR algorithm is guaranteed to converge as it reduces loss (except for the randomness in stochastic gradient descent). The RQ algorithm, on the other hand, has the appealing property that its special case is standard minimization of CE loss on hard labels. QR will likely inherit some VAE properties while RQ some of the behaviours in wake-sleep algorithms; ultimately, though, we find that it really depends on the application, with RQ working across all applications we tried but sometimes being slightly beaten by QR,
  - simple ways to avoid degenerate solutions: For example, when we do not have any hard labels to bootstrap the training, we can still simply start the training with CE loss and then switch to RQ or QR loss; CE training breaks the symmetry initially, and further implicit generative modeling sharpens the predictions.

We believe this section will strengthen a core message of our work -- that implicit generative modeling is a powerful framework for resolving label uncertainties across domains -- while reflecting the practical message that the best design choices may differ slightly for each domain, and that there are still further theoretical studies needed to fully understand how it is possible to avoid explicitly modeling the generative process and rely solely on the inverse modeling. We should note that this was the case with other approaches early on, e.g., addressing bias-variance issues in VAEs and mode collapse in GANs (whose basic formulation allows generating only a single image in the dataset).

- On a similar note, we attempt to compare to the most relevant work across the diverse domains we study in the parts of the text where we introduce and discuss each new domain. However, several reviewers asked for clarification of state of the art across tasks. In light of this feedback, we will incorporate an explicit related work section in the appendix, where we can unify our discussion of related work in each application domain, and in the overall task of learning from probabilistic labels. We believe this addition will strengthen the clarity of our contributions in context of prior work.

For more details on all these points, see responses to individual reviews.

---

### Author Response · Authors · 2021-11-21
**Updated pdf posted**

In response to reviewers' questions and suggestions, we have made changes to the version of our paper available on OpenReview. We highlight the following changes:
- New pseudolabeling result added to Table 1;
- **Practical considerations** section added as Appendix A, discussing batching, loss function selection, and symmetry-breaking concerns;
- **Expanded discussion of related work** added as Appendix B.

---

### Decision · Program_Chairs · 2022-01-20

**Decision:**

Reject

**Comment:**

The authors have addressed several of the issues raised by the reviewers, and they are strongly encouraged in include the additional experiments, and sections, that they propose, in a revised submission. The reviewers also recognized the novelty and extend of applications the proposed methodology has. Nevertheless, the paper would significantly benefit from a rigorous and thorough comparison to related work, placing it well within the context of the literature brought up by reviewers. Experimental comparisons to competitors, even if the latter address more restrictive settings, would strengthen the paper. Most importantly, the authors should consider including a comprehensive related work section, that convincingly discusses and compares to related/adjacent methods.